# PRIMPHORMER: LEVERAGING PRIMAL REPRESENTATION FOR GRAPH TRANSFORMERS

## ABSTRACT

Graph Transformers (GTs) have emerged as a promising approach for graph representation learning. Despite their successes, the quadratic complexity of GTs limits scalability on large graphs due to their pair-wise computations. To fundamentally reduce the computational burden of GTs, we introduce Primphormer, a primal-dual framework that interprets the self-attention mechanism on graphs as a dual representation and then models the corresponding primal representation with linear complexity. Theoretical evaluations demonstrate that Primphormer serves as a universal approximator for functions on both sequences and graphs, showcasing its strong expressive power. Extensive experiments on various graph benchmarks demonstrate that Primphormer achieves competitive empirical results while maintaining a more user-friendly memory and computational costs.

## 1 INTRODUCTION

Graph representation learning has been successfully applied in various fields, including social network analysis (Li et al., 2023), traffic prediction (Dong et al., 2023), and drug discovery (Liu et al., 2023), among others. Much of the research in graph representation learning has focused on Message Passing Neural Networks (MPNNs) which rely on *local* message-passing mechanisms. Although MPNNs have emerged as a powerful approach to short-range tasks that require information exchange among nodes in local neighborhoods, MPNNs face inherent limitations such as over-smoothing (Nguyen et al., 2023), over-squashing (Giraldo et al., 2023), and limited expressivity (Xu et al., 2019; Morris et al., 2019) in long-range tasks (Dwivedi et al., 2022b).

To overcome the limitations, Graph Transformers (GTs) which allow each node to *globally* attend to all other nodes is proposed to enable the learning of long-range dependencies within the graph (Rampasek et al., 2022; Chen et al., 2022). While GT is a promising approach, it has a notable drawback in the *quadratic* complexity, i.e., pair-wise computations in self-attention mechanisms, preventing their practical use.

The key to reducing the quadratic complexity is to use computationally efficient attention mechanisms. Linear attentions like Performer (Choromanski et al., 2021) and BigBird (Zaheer et al., 2020) have been integrated into GTs. However, they need to introduce additional computational overhead, which becomes the dominating source of computation for medium-sized graphs (Rampasek et al., 2022). An alternative approach is sparse attention. Shirzad et al. (2023) introduced Exphormer, a sparse attention mechanism that exchanges information only across edges. The efficiency of Exphormer benefits from the sparsity of graphs. However, its computational complexity increases to quadratic with the number of nodes as graphs become denser, thereby limiting its scalability.

To fundamentally enhance the scalability of GTs, it is crucial to avoid pair-wise computations, prompting us to consider the primal-dual relationship in kernel machines. Examples of models leveraging this relationship include the support vector machine (Cortes & Vapnik, 1995), the least squares support vector machine (Suykens & Vandewalle, 1999), and the kernel principal component analysis (Mika et al., 1999). The primal-dual relationship represents pair-wise symmetric similarity in duality as an inner product of feature mappings in the primal space. By solving optimization problems in the primal space with these feature mappings, quadratic complexity can be avoided.

When constructing the primal representation of the self-attention mechanism, we encounter an essential problem that attention scores are inherently asymmetric, violating the Mercer's condition

(Mercer, 1909), which causes the classical primal-dual discussion to fail. Recent research on primal-dual relationships has sought to explore methods for accommodating asymmetry in kernel machines (Suykens, 2016; He et al., 2023a). In Chen et al. (2023), the self-attention on *sequences* was interpreted through kernel singular value decomposition. This approach collects data information through uniformly sampling the sequence under an *inductive bias* assumption that sequences are ordered. However, this assumption does not hold for graphs, as the graph structure is determined by the edges, and the arrangement or ordering of nodes is not explicitly specified, making it unsuitable for graph-based learning tasks.

**Our contributions.** We propose a novel primal representation for graph Transformers, named *Primphormer*. This method supports asymmetry in self-attention on graphs by introducing an asymmetric kernel trick. It avoids costly pair-wise computations and storage overhead without introducing additional heavy computational burden. The primal-dual analysis reveals that Primphormer can leverage graph information to adjust the basis of outputs, thereby potentially enhancing the model's capacity. Since Primphormer is a new architecture for GTs, we are also interested in its expressive power. To explore this, we prove that Primphormer serves as a universal approximator for arbitrary continuous functions on a compact domain. Through extensive experiments on various graph benchmarks, we show that Primphormer achieves competitive empirical results while maintaining a more user-friendly memory and computational costs.

## 2 METHODS

**Notations.** A graph is denoted as $G = (V, E)$ where $V, E$ are the node and edge sets. $|V| = N$, $|E| = M$ denote the numbers of nodes and edges, respectively. $[N] := \{1, \cdots, N\}$. We take $a, \boldsymbol{a}, \boldsymbol{A}$ to be a scalar, a vector, and a matrix. The inner product of two vectors is written as $\langle \cdot, \cdot \rangle$. The infinite norm of functions is written as $\| \cdot \|_\infty$. The set size is denoted as $| \cdot |$. $\mathbb{R}$ denotes the set of real numbers. $\mathbb{R}_+$ denotes the set of real and positive numbers. $\text{vec}(\boldsymbol{A})$ denotes the vectorization of the matrix $\boldsymbol{A}$, formed by stacking the columns of $\boldsymbol{A}$ into a single column vector. $\otimes$ denotes the Kronecker product. $N_s \ll N$ denotes a small number. $\boldsymbol{1}$ and $\boldsymbol{0}$ denote vectors with all 1 and 0, respectively. $\boldsymbol{X} := [\boldsymbol{x}_1, \cdots, \boldsymbol{x}_N] \in \mathbb{R}^{d \times N}$ is the embedding matrix for nodes where $\boldsymbol{x}_i \in \mathbb{R}^d$ is the embedding of the $i$-th node.

### 2.1 ATTENTION MECHANISM ON GRAPHS

An attention mechanism on a graph $G$ treats nodes $V$ as tokens and is modeled by a fully connected, directed graph that encodes the geometry of $G$ in the positional encoding. Its directed edges denote a directed interaction or similarity between two nodes $i, j$, computed by the inner product in the attention mechanism. Mathematically, we define the attention mechanism as follows,

$$\kappa(\boldsymbol{x}_i, \boldsymbol{x}_j) = \sigma\left(\langle q(\boldsymbol{x}_i), k(\boldsymbol{x}_j) \rangle\right), \quad \boldsymbol{o}_i = \sum_{j=1}^{N} v(\boldsymbol{x}_j) \kappa(\boldsymbol{x}_i, \boldsymbol{x}_j), \quad i, j \in [N], \qquad (2.1)$$

where $\kappa(\boldsymbol{x}_i, \boldsymbol{x}_j)$ is the attention score of node $i$ to node $j$ and $\boldsymbol{o}_i$ is the attention output of vertex $i$. $\sigma$ is an activation function. We denote $q(\boldsymbol{x}) := \boldsymbol{W}_q \boldsymbol{x}, k(\boldsymbol{x}) := \boldsymbol{W}_k \boldsymbol{x}$, and $v(\boldsymbol{x}) := \boldsymbol{W}_v \boldsymbol{x}$ for queries, keys, and values, respectively, and $\boldsymbol{W}_q, \boldsymbol{W}_k, \boldsymbol{W}_v \in \mathbb{R}^{m \times d}$ are learnable weights.

It is worth noting that the attention score is computed for every pair of nodes, leading to memory and computational costs of $\mathcal{O}(N^2)$, which becomes prohibitively expensive for large graphs. Many computationally efficient attention mechanisms are proposed to tackle this issue (Zaheer et al., 2020; Choromanski et al., 2021; Zhuang et al., 2023). Exphormer (Shirzad et al., 2023), a sparse graph transformer, is specifically designed for functions on graphs, which facilitates information exchange across real and expander edges, reducing the memory and computational cost to $\mathcal{O}(N + M)$. However, Exphormer fails its efficiency when dealing with denser graphs, where its computational complexity increases to $\mathcal{O}(N^2)$ as graphs become denser, limiting its scalability.

Such quadratic complexity also exists in kernel machines, where the kernel matrix preserves pair-wise similarities in the dual space. For large-scale problems, it is more practical to contemplate feature representation in the primal space to circumvent quadratic complexity (Fan et al., 2008). One can refer to the representer theorem (Kimeldorf & Wahba, 1971), which delineates the optimal

solution between the primal and dual spaces,

$$g(\boldsymbol{x}_i) = \sum\nolimits_{j=1}^{N} \alpha_j \kappa(\boldsymbol{x}_i, \boldsymbol{x}_j) = \sum\nolimits_{j=1}^{N} \alpha_j \langle \boldsymbol{\phi}(\boldsymbol{x}_i), \boldsymbol{\phi}(\boldsymbol{x}_j) \rangle$$
$$= \langle \boldsymbol{\phi}(\boldsymbol{x}_i), \sum\nolimits_{j=1}^{N} \alpha_j \boldsymbol{\phi}(\boldsymbol{x}_j) \rangle := \langle \boldsymbol{\phi}(\boldsymbol{x}_i), \boldsymbol{w} \rangle, \tag{2.2}$$

where $\alpha_j \in \mathbb{R}$ and $\boldsymbol{w} \in \mathbb{R}^p$ are variables in the dual and primal spaces. $\boldsymbol{\phi} : \mathbb{R}^d \to \mathbb{R}^p$ is the associated feature mapping of the kernel $\kappa$. For vector dual variables $\boldsymbol{\alpha}_j$, we can apply (2.2) to each dimension of $\boldsymbol{\alpha}_j \in \mathbb{R}^s$. Mathematically we have,

$$\tilde{\boldsymbol{g}}(\boldsymbol{x}_i) = \sum_{j=1}^{N} \boldsymbol{\alpha}_j \kappa(\boldsymbol{x}_i, \boldsymbol{x}_j) = \sum_{j=1}^{N} \boldsymbol{\alpha}_j \langle \boldsymbol{\phi}(\boldsymbol{x}_i), \boldsymbol{\phi}(\boldsymbol{x}_j) \rangle = \sum_{j=1}^{N} \mathrm{vec}\left(\boldsymbol{\alpha}_j \boldsymbol{\phi}(\boldsymbol{x}_i)^{\top} \boldsymbol{\phi}(\boldsymbol{x}_j)\right)$$
$$\overset{(a)}{=} \sum_{j=1}^{N} \left(\boldsymbol{\phi}(\boldsymbol{x}_j)^{\top} \otimes \boldsymbol{\alpha}_j\right) \boldsymbol{\phi}(\boldsymbol{x}_i) = \left\langle \sum_{j=1}^{N} \boldsymbol{\phi}(\boldsymbol{x}_j) \otimes \boldsymbol{\alpha}_j^{\top}, \boldsymbol{\phi}(\boldsymbol{x}_i) \right\rangle := \langle \boldsymbol{W}, \boldsymbol{\phi}(\boldsymbol{x}_i) \rangle, \tag{2.3}$$

where $(a)$ comes from the vectorization property of the Kronecker product (Graham, 2018) and $\boldsymbol{W} \in \mathbb{R}^{p \times s}$. The output $\tilde{\boldsymbol{g}}$ in the dual space and the attention output share a similar formulation, indicating that the attention mechanism could potentially be represented in the primal space.

However, the attention score is inherently asymmetric, which violates the Mercer condition (Mercer, 1909). Several works studied this issue and provided a mathematical foundation for allowing asymmetry, as the following definition,

**Definition 1** (Asymmetric kernel trick, (Wright & Gonzalez, 2021; Lin et al., 2022; He et al., 2023a; Chen et al., 2023)). *An asymmetric kernel trick from reproducing kernel Banach spaces (RKBS) with the associated kernel function $\kappa(\cdot, \cdot) : \mathcal{X} \times \mathcal{Z} \to \mathbb{R}$ can be defined by the inner product of two real measurable feature maps from a pair of Banach spaces $\mathcal{B}_{\mathcal{X}}, \mathcal{B}_{\mathcal{Z}}$ on $\mathcal{X}, \mathcal{Z}$:*

$$\kappa(\boldsymbol{x}, \boldsymbol{z}) = \langle \boldsymbol{\phi}_q(\boldsymbol{x}), \boldsymbol{\phi}_k(\boldsymbol{z}) \rangle, \quad \forall \boldsymbol{x} \in \mathcal{X}, \boldsymbol{\phi}_q \in \mathcal{B}_{\mathcal{X}}, \boldsymbol{z} \in \mathcal{Z}, \boldsymbol{\phi}_k \in \mathcal{B}_{\mathcal{Z}}. \tag{2.4}$$

## 2.2 PRIMPHORMER

Here, we elaborate on the construction of Primphormer. A unique characteristic of the aforementioned kernels is their asymmetry, denoted as $\kappa(\boldsymbol{x}, \boldsymbol{y}) \neq \kappa(\boldsymbol{y}, \boldsymbol{x})$. This can be understood as a directional similarity from a query to a key, providing a pair of directed similarities between $\boldsymbol{x}, \boldsymbol{y}$. Consequently, for each input $\boldsymbol{x}$, the output should be computed by considering aspects of both queries and keys: $\boldsymbol{e}(\boldsymbol{x}) := \sum_j \boldsymbol{h}_j \kappa(\boldsymbol{x}, \boldsymbol{x}_j)$ and $\boldsymbol{r}(\boldsymbol{x}) := \sum_i \boldsymbol{h}_i \kappa(\boldsymbol{x}_i, \boldsymbol{x})$. It is intriguing to investigate a suitable primal representation, as we recognize the resemblance in formulation between attention outputs and the dual representation in kernel machines, both associated with an asymmetric kernel. To address this, we present an optimization problem to explore its primal-dual relationship,

$$\min_{\boldsymbol{W}_e, \boldsymbol{W}_r, \boldsymbol{e}_i, \boldsymbol{r}_j} J = \frac{1}{2} \sum_{i=1}^{N} \boldsymbol{e}_i^{\top} \boldsymbol{\Lambda} \boldsymbol{e}_i + \frac{1}{2} \sum_{j=1}^{N} \boldsymbol{r}_j^{\top} \boldsymbol{\Lambda} \boldsymbol{r}_j - \mathrm{Tr}(\boldsymbol{W}_e^{\top} \boldsymbol{W}_r)$$
$$\text{s.t.} \quad \boldsymbol{e}_i = f_X \boldsymbol{W}_e \boldsymbol{\phi}_q(\boldsymbol{x}_i), i \in [N],$$
$$\boldsymbol{r}_j = f_X \boldsymbol{W}_r \boldsymbol{\phi}_k(\boldsymbol{x}_j), j \in [N], \tag{2.5}$$

where $\boldsymbol{W}_e, \boldsymbol{W}_r \in \mathbb{R}^{N_s \times p}$ are the projection weights, $\boldsymbol{\Lambda} \in \mathbb{R}_+^{s \times s}$ represents a diagonal regularization coefficient matrix. $\boldsymbol{\phi}_q(\cdot), \boldsymbol{\phi}_k(\cdot) : \mathbb{R}^d \to \mathbb{R}^p$ correspond to the feature maps of queries and keys, respectively. The expected primal representations are the projection scores $\boldsymbol{e}_i, \boldsymbol{r}_j \in \mathbb{R}^s$ in the constraints. $f_X \in \mathbb{R}^{s \times N_s}$ is a data-dependent projection and is defined by $f_X := \boldsymbol{F} + \boldsymbol{B} \boldsymbol{X} \mathbf{1}_s \mathbf{1}_{N_s}^{\top}$ with data-independent projections $\boldsymbol{F} \in \mathbb{R}^{s \times N_s}$ and $\boldsymbol{B} \in \mathbb{R}^{s \times d}$. In graph representation learning, $f_X$ serves as a virtual node (Cai et al., 2023) that aggregates information of each node in the graph.

The objective function $J$ minimizes the coupling term and the squares of $\boldsymbol{e}, \boldsymbol{r}$ regarding queries and keys by introducing a variational principle of asymmetric kernels as discussed by Suykens (2016). Below, we present the theorem on the solution to the dual problem of the primal problem (2.5),

**Theorem 1** (Duality of the optimization (2.5)). *The dual problem of the optimization (2.5) under the Karush-Kuhn-Tucker (KKT) conditions is the following linear system,*

$$\boldsymbol{K}\boldsymbol{H}_r\boldsymbol{F}_X = \boldsymbol{H}_e\boldsymbol{\Sigma},$$
$$\boldsymbol{K}^\top\boldsymbol{H}_e\boldsymbol{F}_X = \boldsymbol{H}_r\boldsymbol{\Sigma},$$

(2.6)

*which collects the solutions corresponding to the non-zero entries in $\boldsymbol{\Lambda}$ such that $\boldsymbol{\Sigma} := \boldsymbol{\Lambda}^{-1}$. $\boldsymbol{H}_e := [\boldsymbol{h}_{e_1}, \dots, \boldsymbol{h}_{e_N}]^\top \in \mathbb{R}^{N \times s}$, and $\boldsymbol{H}_r := [\boldsymbol{h}_{r_1}, \dots, \boldsymbol{h}_{r_N}]^\top \in \mathbb{R}^{N \times s}$ are dual variables. $\boldsymbol{K}$ corresponds to the attention score, induced by $\boldsymbol{K}_{ij} := \langle \boldsymbol{\phi}_q(\boldsymbol{x}_i), \boldsymbol{\phi}_k(\boldsymbol{x}_j) \rangle$. The detailed proofs, Lagrangian, and KKT conditions are provided in Appendix C.1.*

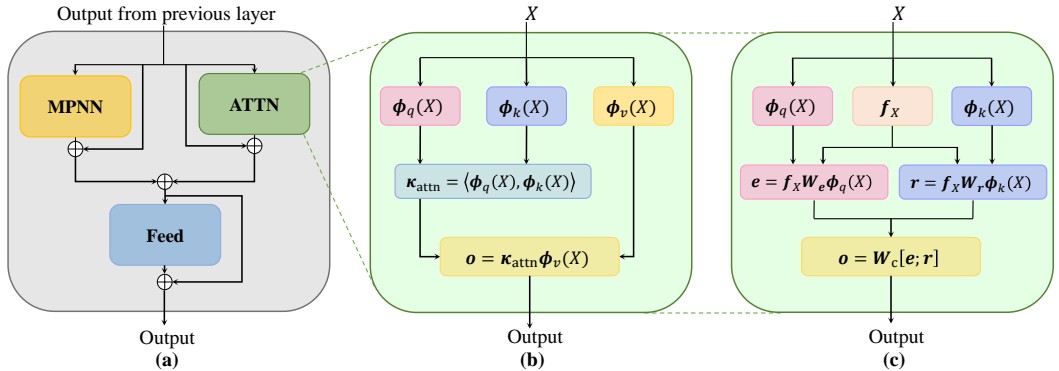

Figure 1 Illustrations of the architectures in one layer. **a)** The GPS architecture. **b)** The standard self-attention architecture. The attention score $\kappa_{\mathrm{attn}}$ is induced by two feature mappings $\phi_q$ and $\phi_k$ involving pair-wise computations. **c)** Primphormer eliminates the need for pair-wise computations by introducing the primal representation, resulting in a new computationally efficient GT.

**Primal and dual relationship.** The KKT conditions (C2) yields a fact that the optimized projections $\boldsymbol{W}_r$ and $\boldsymbol{W}_e$ in the primal space are composed of all the tokens,

$$\begin{cases} \boldsymbol{W}_e = \sum_{j=1}^{N} f_X^\top \boldsymbol{h}_{r_j} \boldsymbol{\phi}_k(\boldsymbol{x}_j)^\top, \\ \boldsymbol{W}_r = \sum_{i=1}^{N} f_X^\top \boldsymbol{h}_{e_i} \boldsymbol{\phi}_q(\boldsymbol{x}_i)^\top. \end{cases}$$

(2.7)

According to the primal-dual relationship between (2.5) and (2.6), and by applying (2.7) to the projection scores $\boldsymbol{e}, \boldsymbol{r}$, we can formulate them in the following two ways: (a) the primal representation under KKT conditions, and (b) the dual representation as the standard self-attention mechanism,

$$\text{Primal}: \begin{cases} \boldsymbol{e}(\boldsymbol{x}) = f_X \boldsymbol{W}_e \boldsymbol{\phi}_q(\boldsymbol{x}), \\ \boldsymbol{r}(\boldsymbol{x}) = f_X \boldsymbol{W}_r \boldsymbol{\phi}_k(\boldsymbol{x}), \end{cases} \quad \text{Dual}: \begin{cases} \boldsymbol{e}(\boldsymbol{x}) = \sum_{j=1}^{N} \tilde{\boldsymbol{h}}_{r_j} \kappa(\boldsymbol{x}, \boldsymbol{x}_j), \\ \boldsymbol{r}(\boldsymbol{x}) = \sum_{i=1}^{N} \tilde{\boldsymbol{h}}_{e_i} \kappa(\boldsymbol{x}_i, \boldsymbol{x}), \end{cases}$$

(2.8)

where $\boldsymbol{F}_X := f_X f_X^\top$, and $\tilde{\boldsymbol{h}}_{r_j} := \boldsymbol{F}_X \boldsymbol{h}_{r_j}, \tilde{\boldsymbol{h}}_{e_i} := \boldsymbol{F}_X \boldsymbol{h}_{e_i}$. In the primal space, we integrate token information into the projection weights $\boldsymbol{W}_r$ and $\boldsymbol{W}_e$ (2.7), representing self-attention through linear projection to avoid pair-wise computations. The data-dependent projection $f_X$ inside serves as a virtual node aggregating information across all graph nodes, intended to introduce graph information to each node. Correspondingly, in the dual space, the attention score is computed using an asymmetric kernel trick, denoted as $\kappa(\boldsymbol{x}_i, \boldsymbol{x}_j) := \langle \boldsymbol{\phi}_q(\boldsymbol{x}_i), \boldsymbol{\phi}_k(\boldsymbol{x}_j) \rangle$, and the data-adaptive basis $\tilde{\boldsymbol{h}}_{r_j}, \tilde{\boldsymbol{h}}_{e_i}$ act as values, forming a standard self-attention formulation. These values are influenced by $f_X$, leading to an auto-correlation projection $\boldsymbol{F}_X$ without centering $f_X$. This auto-correlation projection $\boldsymbol{F}_X$ is significantly affected by the mean value of $f_X$, i.e., the virtual node. This offers a new perspective on bridging virtual nodes and the self-attention mechanism.

Chen et al. (2023) introduced an alternative form of primal-dual relationship for sequence data. Its data-dependent projection is uniformly sampled from sequences under an inductive bias assumption that sequences are ordered, which is natural to sequences but not graphs. Sequences are inherently

ordered, and thus such sub-sequences contain semantic information from the original sequence. In contrast, for graph data, their structure is dictated by the edges, and the arrangement or ordering of nodes is not explicitly specified, rendering this method unsuitable for graph data. Moreover, its data-dependent projection is integrated into the kernel trick as a data-adaptive weight, incapable of altering the space where potential outputs may lie. In contrast, our data-adaptive basis aggregates graph information in the form of virtual nodes and directly influences the basis of outputs, as shown in equation (2.8), potentially enhancing the model's capacity.

**Model architecture.** The Transformer layer consists of two core components: the self-attention module and the feed-forward module which is applied token-wise (Vaswani et al., 2017). In this paper, we consider GPS, a powerful GT architecture that merges the MPNN and Transformer layers (Rampasek et al., 2022). We replace the self-attention module in the Transformer layer with our primal representation and name our method Primphormer. Illustrations of Primphormer's architecture are shown in Figure 1, with detailed algorithms presented in Appendix D.

**Complexity analysis.** The primal representation is a more user-friendly approach in terms of both time and memory costs. The dual representation requires $\mathcal{O}(N^2 s)$ time complexity and $\mathcal{O}(N^2 + Ns)$ memory complexity. In contrast, the primal representation only requires $\mathcal{O}(Nps)$ time complexity and $\mathcal{O}(2N_s s + 2Np)$ memory complexity with $N_s \ll N$ making an efficient self-attention mechanism feasible. The final output is obtained by concatenating two projection scores $\boldsymbol{o}(\boldsymbol{x}) := [\boldsymbol{e}(\boldsymbol{x}); \boldsymbol{r}(\boldsymbol{x})]$. To align with the user-dependent dimension $d_\text{o}$, a compatibility matrix $\boldsymbol{W}_c \in \mathbb{R}^{d_\text{o} \times 2s}$ can be further applied to the output score.

## 3 THEORETICAL RESULTS

In this section, we provide the main theorems of Primphormer. The proof details can be found in Appendix C.

### 3.1 ZERO-VALUED OBJECTIVE

In the implementation of Primphormer, our goal is to reach the KKT point. Theorem 1 establishes that when the KKT conditions are met, the dual representation of Primphormer aligns with the standard self-attention formulation. However, solving the linear system (2.6) in the dual space introduces a cubic computational complexity. To efficiently approach the KKT points, we introduce the following theorem,

**Theorem 2** (Zero-valued objective with stationary solutions). *The solutions of $\boldsymbol{H}_e, \boldsymbol{H}_r, \boldsymbol{\Sigma}$ in the dual space (2.6) lead to a zero-valued objective $J$ in the primal space (2.5).*

The essence of Theorem 2 lies in the necessity for the primal objective value to be zero under the KKT conditions, suggesting an alternative optimization approach instead of solving the dual problem. Therefore, we implement Primphormer by jointly minimizing an additional loss towards zero as follows,

$$\mathcal{L} = \mathcal{L}_\text{task} + \eta \sum_l J_l^2, \tag{3.1}$$

where $\eta \in \mathbb{R}_+$ is a regularization coefficient, $\mathcal{L}_\text{task}$ is the task-oriented loss and the final term sums up the primal objective loss (2.5) across layer $l$. Through regularization of this additional loss, the self-attention mechanism can be effectively represented in the primal space upon achieving a zero-valued objective.

### 3.2 UNIVERSAL APPROXIMATION

By substituting the self-attention layer with our primal representation, we obtain a new network architecture. Subsequently, the first question that intrigues us concerns expressivity, particularly delving into which functions can be uniformly approximated utilizing our network. Here, we demonstrate that Primphormer allows universal approximation for continuous functions on both sequences and graphs. The proofs of these theorems rely on a mild assumption: let feature spaces be $\mathcal{X}, \mathcal{Y} \subseteq \mathbb{R}^d$ and let $\mathcal{X}$ be a compact set. We first introduce the concept of permutation equivariance and then show that Primphormer is a universal approximator.

**Definition 2** (Permutation equivariance, (Hutter, 2020; Alberti et al., 2023)). *A continuous sequence-to-sequence function $f : \mathcal{X}^N \to \mathcal{Y}^N$ is equivariant to the order of elements in a sequence if for each permutation $\pi : [N] \to [N]$,*

$$f\left(\left[\boldsymbol{x}_{\pi(1)}, \cdots, \boldsymbol{x}_{\pi(N)}\right]\right) = \left[f_{\pi(1)}(\boldsymbol{X}), \cdots, f_{\pi(N)}(\boldsymbol{X})\right],$$

*where $\mathcal{X}^N \ni \boldsymbol{X} = [\boldsymbol{x}_1, \cdots, \boldsymbol{x}_N]$ is a sequence of N elements. We denote $f \in \mathcal{F}_{\mathrm{eq}}^N(\mathcal{X}, \mathcal{Y})$ if $f$ conforms to this definition.*

We are now ready to state the universal approximation property of Primphormer on permutation equivariant sequence-to-sequence functions.

**Theorem 3.** *For any function $f \in \mathcal{F}_{\mathrm{eq}}^N(\mathcal{X}, \mathcal{Y})$ and for each $\epsilon > 0$ there exists a Primphormer $\mathcal{T}_{\mathrm{Pri}}$ such that*

$$\sup_{\boldsymbol{X} \in \mathcal{X}^N} \|f(\boldsymbol{X}) - \mathcal{T}_{\mathrm{Pri}}(\boldsymbol{X})\|_\infty < \epsilon. \tag{3.2}$$

Next, we develop the theorem for any continuous sequence-to-sequence function, stating that with a positional encoding $\boldsymbol{E} \in \mathbb{R}^{d \times N}$, a Primphormer $\mathcal{T}_{\mathrm{PE}}(\boldsymbol{X}) = \mathcal{T}_{\mathrm{Pri}}(\boldsymbol{X} + \boldsymbol{E})$ can approximate any continuous sequence-to-sequence functions on the compact domain.

**Theorem 4.** *For any continuous function $f : [0,1]^{d \times N} \to \mathbb{R}^{d \times N}$ and for each $\epsilon > 0$ there exists a Primphormer with the positional encoding $\mathcal{T}_{\mathrm{PE}}$ such that*

$$\sup_{\boldsymbol{X} \in \mathcal{X}^N} \|f(\boldsymbol{X}) - \mathcal{T}_{\mathrm{PE}}(\boldsymbol{X})\|_\infty < \epsilon. \tag{3.3}$$

Theorems 3, 4 provide universal approximation properties for functions on *sequences*. In the realm of graph learning, an interesting question arises: does the universality extend to functions on *graphs*?

**Universal approximator for functions on graphs**. To answer the question, we construct node and edge Primphormers on graphs. For an input graph $G$, the edge Primphormer processes input as a sequence of ordered pairs $((i, j), \sigma_{ij})$ where $i \leq j$, $i, j \in [N]$ and an edge indicator $\sigma_{ij}$. It is evident that any permutation on these pairs describes the same graph. Considering the set of functions $f : \mathbb{R}^{N \times (N-1)} \to \mathbb{R}^{N \times (N-1)}$ with permutation equivariance, Theorem 3 asserts that the function $f$ can be approximated with arbitrary accuracy by Primphormer on edge input. Similarly, the node Primphormer takes an identity matrix as input and the padded adjacency matrix as a positional encoding which can be interpreted as a one-hot encoding of each node's neighbors. Considering the set of continuous functions $f : [0,1]^{N \times N} \to \mathbb{R}^{N \times N}$, Theorem 4 states that $f$ can be approximated as closely as desired by an appropriate Primphormer on node inputs. These results indicate that Primphormer can offer an approximate solution to the graph isomorphism problem, although they do not imply the existence of efficient algorithms for solving this problem. For more detailed explorations, we recommend referring to Kreuzer et al. (2021).

## 4 EXPERIMENTAL RESULTS

In this section, we evaluate the empirical performance of Primphormer on various graph benchmarks. To ensure diversity, datasets are collected from different sources, a detailed description of which can be found in Appendix A. In particular, we conducted experiments on the benchmark datasets including the image-based graph datasets CIFAR10, MNIST, COCO-SP, and PascalVOC-SP; the synthetic SBM datasets PATTERN and CLUSTER; the code graph dataset MalNet-Tiny; the molecular datasets including Peptides-Func, Peptides-Struct, and PCQM-Contact (Dwivedi et al., 2022a; Freitas et al., 2021; Dwivedi et al., 2022b; 2023); and the large-scale ogbn-products dataset (Hu et al., 2020). In our experiments, we use feature maps defined as $\phi_q(\boldsymbol{x}) := q(\boldsymbol{x})/\|q(\boldsymbol{x})\|_2$ and $\phi_k(\boldsymbol{x}) := k(\boldsymbol{x})/\|k(\boldsymbol{x})\|_2$ as used by Chen et al. (2023).

**Long-range graph benchmark.** We conducted experiments on the long-range graph benchmark (LRGB, Dwivedi et al. (2022b)) to evaluate the models' capabilities in learning long-range dependencies within input graphs. Table 1 presents the results of Primphormer with several baselines. Our approach outperforms the baselines on three of the five datasets while showing competitive performance on the rest of the datasets.

**GNN benchmark datasets.** We also evaluate our method with broader baselines on graph benchmark datasets, namely CIFAR10, MNIST, CLUSTER, PATTERN, and the code graph dataset MalNet-Tiny (Dwivedi et al., 2023; Freitas et al., 2021), as reported in Table 2. It is observed that Primphormer outperforms on MNIST and ranks as the second-best performer on two additional datasets, showcasing its strong performance across various dataset types.

Table 1 Comparison of Primphormer with baselines on the long-range graph benchmark. Best results are colored in first, second, third.

| Model | PascalVOC-SP F1↑ | COCO-SP F1↑ | Peptides-Func AP↑ | Peptides-Struct MAE↓ | PCQM-Contact MRR↑ |
|---|---|---|---|---|---|
| GCN | 0.1268 ± 0.0060 | 0.0841 ± 0.0010 | 0.5930 ± 0.0023 | 0.3496 ± 0.0013 | 0.3234 ± 0.0006 |
| GINE | 0.1265 ± 0.0076 | 0.1339 ± 0.0044 | 0.5498 ± 0.0079 | 0.3547 ± 0.0045 | 0.3180 ± 0.0027 |
| GatedGCN | 0.2873 ± 0.0219 | 0.2641 ± 0.0045 | 0.5864 ± 0.0077 | 0.3420 ± 0.0013 | 0.3218 ± 0.0011 |
| GatedGCN+RWSE | 0.2860 ± 0.0085 | 0.2574 ± 0.0034 | 0.6069 ± 0.0035 | 0.3357 ± 0.0006 | 0.3242 ± 0.0008 |
| Trans.+LapPE | 0.2694 ± 0.0098 | 0.2618 ± 0.0031 | 0.6326 ± 0.0126 | 0.2529 ± 0.0016 | 0.3174 ± 0.0020 |
| SAN+LapPE | 0.3230 ± 0.0039 | 0.2592 ± 0.0158 | 0.6384 ± 0.0121 | 0.2683 ± 0.0043 | 0.3350 ± 0.0003 |
| SAN+RWSE | 0.3216 ± 0.0027 | 0.2434 ± 0.0156 | 0.6439 ± 0.0075 | 0.2545 ± 0.0012 | 0.3341 ± 0.0006 |
| GraphGPS | 0.3748 ± 0.0109 | 0.3412 ± 0.0044 | 0.6535 ± 0.0041 | 0.2500 ± 0.0005 | 0.3337 ± 0.0006 |
| Exphormer | 0.3975 ± 0.0037 | 0.3455 ± 0.0009 | 0.6527 ± 0.0043 | 0.2481 ± 0.0007 | 0.3637 ± 0.0020 |
| Primphormer | 0.3980 ± 0.0075 | 0.3438 ± 0.0046 | 0.6612 ± 0.0065 | 0.2495 ± 0.0008 | 0.3757 ± 0.0079 |

Table 2 Comparison of Primphormer with baselines on GNN benchmark datasets. Best results are colored in first, second, third.

| Model | CIFAR10 Accuracy↑ | MalNet-Tiny Accuracy↑ | MNIST Accuracy↑ | CLUSTER Accuracy↑ | PATTERN Accuracy↑ |
|---|---|---|---|---|---|
| GCN | 55.71 ± 0.381 | 81.0 | 90.71 ± 0.218 | 68.50 ± 0.976 | 71.89 ± 0.334 |
| GIN | 55.26 ± 1.527 | 88.98 ± 0.557 | 96.49 ± 0.252 | 64.72 ± 1.553 | 85.39 ± 0.136 |
| GAT | 64.22 ± 0.455 | 92.10 ± 0.242 | 95.54 ± 0.205 | 70.59 ± 0.447 | 78.27 ± 0.186 |
| GatedGCN | 67.31 ± 0.311 | 92.23 ± 0.650 | 97.34 ± 0.143 | 73.84 ± 0.326 | 85.57 ± 0.088 |
| PNA | 70.35 ± 0.630 | - | 97.94 ± 0.120 | - | - |
| DGN | 72.84 ± 0.417 | - | - | - | 86.68 ± 0.034 |
| CRaWL | 69.01 ± 0.259 | - | 97.94 ± 0.050 | - | - |
| GIN-AK+ | 72.19 ± 0.130 | - | - | - | 86.85 ± 0.057 |
| SAN | - | - | - | 76.69 ± 0.650 | 86.58 ± 0.037 |
| K-Subgraph SAT | - | - | - | 77.86 ± 0.104 | 86.85 ± 0.037 |
| EGT | 68.70 ± 0.409 | - | 98.17 ± 0.087 | 79.23 ± 0.348 | 86.82 ± 0.020 |
| GraphGPS | 72.30 ± 0.356 | 93.50 ± 0.410 | 98.05 ± 0.126 | 78.02 ± 0.180 | 86.69 ± 0.059 |
| Exphormer | 74.69 ± 0.125 | 94.02 ± 0.209 | 98.55 ± 0.039 | 78.07 ± 0.037 | 86.74 ± 0.015 |
| Primphormer | 74.13 ± 0.241 | 93.62 ± 0.242 | 98.56 ± 0.042 | 78.01 ± 0.162 | 86.68 ± 0.056 |

**Efficiency validation.** Primphormer leverages the primal representation for GTs to reduce computational burden. As the aforementioned results demonstrate the promising performance of Primphormer, we further validate its efficiency by comparing it to other computationally efficient attention mechanisms within the GPS architecture (Rampasek et al., 2022). The selected mechanisms include linear attention models BigBird (Zaheer et al., 2020) and Performer (Choromanski et al., 2021), a sparse attention mechanism, Exphormer (Shirzad et al., 2023), the sequence-specific Primal-Atten (Chen et al., 2023), and the full attention mechanism. We conduct the experiments on CIFAR10, MalNet-Tiny, PascalVOC, Peptides-Func and a large-scale graph ogbn-products. Since ogbn-products is too large to be loaded into GPU, we use the random partitioning method previously used by Wu et al. (2022; 2023). The results across the five datasets are reported in Tables 3 and 4.

As shown in Table 3, Primphormer demonstrates superior performance over other attention mechanisms such as BigBird, Performer, and Prim-Atten, while also exhibiting competitive performance with Exphormer. Table 4 presents a comparison of running time and peak memory usage across different methods. Primphormer demonstrates superior performance in both running time and memory consumption compared to other approaches. For example, in the MalNet-Tiny dataset, linear attention mechanisms introduce significant computational overhead. While Prim-Atten offers good efficiency, its performance on graph tasks lags due to its sequence-specific nature. Both Primphormer

Table 3 Comparison of attention mechanisms in GPS. Best results are colored in first, second, third. OOM means out of memory.

| Model GPS | CIFAR10 Accuracy↑ | MalNet-Tiny Accuracy↑ | PascalVOC-SP F1↑ | Peptides-Func AP↑ | OGBN-products Accuracy↑ |
|---|---|---|---|---|---|
| MPNN-only | 69.95 ± 0.499 | 92.23 ± 0.650 | 0.3016 ± 0.0031 | 0.6159 ± 0.0048 | 74.25 ± 0.214s |
| +Transformer | 72.31 ± 0.344 | 93.50 ± 0.410 | 0.3736 ± 0.0158 | 0.6535 ± 0.0041 | OOM |
| +BigBird | 70.48 ± 0.106 | 92.34 ± 0.340 | 0.2762 ± 0.0069 | 0.5854 ± 0.0079 | 73.82 ± 0.412 |
| +Performer | 70.67 ± 0.338 | 92.64 ± 0.780 | 0.3724 ± 0.0131 | 0.6475 ± 0.0056 | 74.30 ± 0.211 |
| +Prim-Atten | 71.57 ± 0.256 | 92.97 ± 0.228 | 0.3173 ± 0.0055 | 0.6447 ± 0.0046 | 74.47 ± 0.134 |
| +Exphormer | 74.69 ± 0.125 | 94.02 ± 0.209 | 0.3975 ± 0.0037 | 0.6527 ± 0.0043 | 74.67 ± 0.179 |
| +Primphormer | 74.13 ± 0.241 | 93.62 ± 0.242 | 0.3980 ± 0.0075 | 0.6612 ± 0.0065 | 74.89 ± 0.281 |

Table 4 Efficiency comparisons on running time and peak memory consumption.

| Model GPS | Time (s/epoch) | | | | | Peak memory usage (GB) | | | | |
|---|---|---|---|---|---|---|---|---|---|---|
| | CIFAR. | MalNet. | Pascal. | Func. | prod. | CIFAR. | MalNet. | Pascal. | Func. | prod. |
| MPNN-only | 20.3 | 24.5 | 15.7 | 4.8 | 21.1 | 2.31 | 1.92 | 4.18 | 2.45 | 11.97 |
| +Transformer | 28.0 | 232.4 | 35.6 | 12.8 | - | 3.81 | 35.32 | 7.82 | 8.46 | OOM |
| +BigBird | 55.2 | 325.6 | 52.3 | 51.9 | 93.9 | 2.81 | 2.71 | 4.99 | 4.99 | 17.29 |
| +Performer | 50.8 | 73.5 | 49.7 | 21.7 | 22.7 | 10.5 | 11.59 | 6.14 | 7.71 | 16.14 |
| +Prim-Atten | 32.1 | 62.5 | 25.7 | 7.9 | 22.6 | 2.74 | 2.58 | 4.74 | 3.38 | 13.63 |
| +Exphormer | 44.5 | 62.1 | 35.2 | 7.6 | 25.4 | 5.54 | 10.38 | 7.35 | 4.81 | 31.09 |
| +Primphormer | 32.6 | 61.9 | 25.3 | 7.7 | 22.1 | 2.74 | 2.86 | 4.72 | 3.41 | 13.35 |

and Exphormer, designed for graphs, exhibit similar running times. Nevertheless, Primphormer consumes less memory as its complexity depends solely on the number of nodes, whereas Exphormer's complexity is controlled by the number of nodes and edges. In the ogbn-products dataset, which comprises approximately 2 million nodes and 61 million edges, Primphormer showcases the most efficient results compared with other methods. In summary, our experiments demonstrate that Primphormer exhibits competitive performance while maintaining user-friendly memory and computational costs.

## 5 RELATED WORK

**Graph Transformers.** Transformers have demonstrated success in natural language processing (Vaswani et al., 2017) and computer vision tasks (Liu et al., 2021). Recently, researchers have explored the application of Transformers in graph representation learning to address issues such as over-smoothing (Nguyen et al., 2023) and over-squashing (Giraldo et al., 2023) observed in MPNNs. Graph Transformers operate on a fully connected graph where nodes are pairwise connected, encoding the original graph structure into positional encodings. Spectral Attention Networks (SAN) (Kreuzer et al., 2021) introduce conditional attention for both real and virtual edges and implement Laplacian positional encoding for nodes. Graphormer (Ying et al., 2021) and GraphiT (Mialon et al., 2021) incorporate relative positional encodings based on pairwise graph distances and diffusion kernels, respectively. GPS proposes a framework that combines MPNNs with attention mechanisms (Rampasek et al., 2022).

The quadratic complexity in traditional GTs has motivated the development of computationally efficient attention mechanisms. Nodeformer (Wu et al., 2022) utilizes the kernelized Gumbel softmax operator to facilitate information propagation between all pairs of nodes efficiently. Difformer (Wu et al., 2023) introduces a diffusion-based Transformer model with linear complexity, although their attention mechanisms are limited to nodes in randomly sampled mini-batches. Another strategy is the sparse Transformer, which enhances computational efficiency by restricting node interactions. Exphormer (Shirzad et al., 2023) limits interactions across real and expander edges, achieving linear complexity to the number of nodes and edges. However, the efficiency of Exphormer diminishes as graphs become denser. A survey on efficient Transformers is given by Fournier et al. (2023).

**Primal-dual relationship.** The quadratic complexity also arises in kernel machines in duality and can be circumvented by transferring a dual problem to its primal form. Models such as the support vector machine (Cortes & Vapnik, 1995), least squares support vector machine (Suykens & Vandewalle, 1999), and kernel principal component analysis (Mika et al., 1999) exhibit this characteristic. The associated pair-wise kernels are symmetric and positive-definite, whereas attention scores are inherently asymmetric, violating the Mercer condition (Mercer, 1909). Recent research has explored a new primal-dual perspective to accommodate such asymmetry in kernel machines. To incorporate asymmetric kernel functions, Lin et al. (2022) propose an asymmetric kernel trick from a pair of RKBSs. He et al. (2023b) convert an asymmetric kernel to a complex-valued Hermitian function by the magnetic transform. Suykens (2016) introduces a novel variational principle to dissect the primal-dual relationship concerning the singular value decomposition of an asymmetric kernel matrix, a concept further extended to classification tasks by He et al. (2023a). This variational principle is also leveraged by Chen et al. (2023) to interpret attention mechanisms in sequences. However, due to the distinctions between sequences and graphs, this model is unsuitable for graph-based learning.

## 6    CONCLUSION

In this paper, we propose Primphormer, a new framework for graph Transformers. Primphormer models the self-attention mechanism on graphs in the primal space, avoiding pair-wise computations, which enables an efficient variant of graph Transformers. Our primal-dual analysis shows that Primphormer can be implemented by introducing an additional primal objective loss. Due to its efficiency in both runtime and memory storage, Primphormer has the potential to support larger and deeper neural networks and enable larger batch sizes, enhancing model capacity and generalization ability. Primphormer also benefits from the universal approximation property for functions on both sequences and graphs, potentially possessing strong generalization capabilities to unseen data or tasks. Experimental results on various graph benchmarks demonstrate the effectiveness and efficiency of the proposed Primphormer.

An interesting avenue for future work is exploring how edge features can be incorporated into Primphormer's structure. Edge features can be added to attention scores in an entry-wise manner as data-adaptive kernels (Liu et al., 2020). Exploring the primal representation of these kernels allows us to incorporate edge information into attention mechanisms, potentially resulting in a stronger GT. Additionally, fine-tuning schemes like LoRA (Hu et al., 2022) are promising for large models. Studying LoRA from a primal-dual perspective may lead to more efficient fine-tuning methods. Although experimentally evaluating the universal approximation property poses challenges, it is crucial and valuable for our theoretical foundations. We can begin by manually designing graph-to-graph functions and then study validation errors concerning hidden dimensions and sample size. For tasks that focus on short-range interactions, the data-dependent projection could be further adjusted to better aggregate local information.

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

# APPENDIX

## A   DATA DESCRIPTIONS

Here, we introduce the datasets in the experiments. A summary of the dataset statistics is shown in Tab. A1.

**CIFAR10 and MNIST.** CIFAR10 and MNIST are the graph equivalents of the image classification datasets of the same name. A graph is created by constructing the 8-nearest neighbor graph of the SLIC superpixels of the image. These are both 10-class graph classification problems (Dwivedi et al., 2023).

**PascalVOC-SP and COCO-SP.** These are similar graph versions of image datasets, but they are larger images and the task is to perform node classification, i.e., semantic segmentation of super-pixels. These graphs are larger, and the tasks are more complex than CIFAR10 and MNIST (Dwivedi et al., 2022a).

**CLUSTER and PATTERN.** PATTERN and CLUSTER are node classification problems. Both are synthetic datasets that are sampled from a Stochastic Block Model (SBM), is a popular way to model communities. In PATTERN, the prediction task is to identify if a node belongs to one of the 100 possible predetermined sub-graph patterns. In CLUSTER, the goal is to classify nodes into six different clusters with the same distribution (Dwivedi et al., 2023).

**MalNet-Tiny.** Malnet-Tiny is a smaller dataset generated from a larger dataset for identifying malware based on function call graphs from Android APKs. The tiny dataset contains 5000 graphs, each with up to 5000 nodes. The task is to predict the graph as being benign or from one of four types of malware (Freitas et al., 2021).

**Peptides-Func, Peptides-Struct, and PCQM-Contact.** These datasets are molecular graphs introduced as a part of the Long Range Graph Benchmark (LRGB). On PCQM-Contact, the task is edge-level, and we need to rank the edges. Peptides-Func is a multi-label graph classification task with 10 labels. Peptides-Struct is graph-level regression of 11 structural properties of the molecules (Dwivedi et al., 2022a;b).

**OGBN-products.** The ogbn-products dataset is an undirected and unweighted graph, representing an Amazon product co-purchasing network. Nodes represent products sold in Amazon, and edges between two products indicate that the products are purchased together. Specifically, node features are generated by extracting bag-of-words features from the product descriptions followed by a Principal Component Analysis to reduce the dimension to 100. The task is to predict the category of a product in a multi-class classification setup, where the 47 top-level categories are used for target labels (Hu et al., 2020). We use the random partitioning method with ten partitions as previously utilized in Wu et al. (2022; 2023).

Table A1 Dataset statistics

| Dataset | Graphs | Avg. nodes | Avg.edges | Task level | Class | Metric |
|---|---|---|---|---|---|---|
| MNIST | 70,000 | 70.6 | 564.5 | graph | 10 | Acc |
| CIFAR10 | 60,000 | 117.6 | 941.1 | graph | 10 | Acc |
| PATTERN | 14,000 | 118.9 | 3039.3 | inductive node | 2 | Acc |
| CLUSTER | 12,000 | 117.2 | 2150.9 | inductive node | 6 | Acc |
| MalNet-Tiny | 5,000 | 1,410.3 | 2,859.9 | graph | 5 | Acc |
| PascalVOC-SP | 11,355 | 479.4 | 2710.5 | inductive node | 21 | F1 |
| COCO-SP | 123,286 | 476.9 | 2710.5 | inductive node | 81 | F1 |
| PCQM-Contact | 529,434 | 30.1 | 61.0 | inductive link | link ranking | MRR |
| Peptides-func | 15,535 | 150.9 | 307.3 | graph | 10 | AP |
| Peptides-struct | 15,535 | 150.9 | 309.3 | graph | 11 | MAE |
| OGBN-products | 1 | 2,449,029 | 61,859,140 | node | 47 | Acc |

## B  HYPERPARAMETERS

Our selection of hyperparameters was guided by the instructions in GPS (Rampasek et al., 2022) and Exphormer (Shirzad et al., 2023). Further details can be found in Tables. A3- A4.

In our model, we introduced additional hyperparameters, the dimensions of the data-dependent projection, denoted as $N_s$ and its low rank $s$, and the regularization coefficient $\eta$. We utilized grid search to explore these hyperparameters across $N_s, s \in \{20, 30, 40, 50, 60\}$, and $\eta \in \{0.1, 0.01\}$. For the remaining hyperparameters, we conducted a linear search for each parameter to determine the best values. Throughout all experiments, we employed CustomGatedGCN as the MPNN module alongside Primphormer except for ogbn-products dataset where we use GCN. To ensure fair comparisons, we maintained a similar parameter budget to that of GraphGPS.

Table A4 presents the hyperparameters used in our efficiency experiments. To maintain consistency in our evaluations of various attention mechanisms, we applied the same parameters for a fair comparison.

Table A2 Hyperparameters used in Primphormer for datasets: PascalVOC-SP, COCO-SP, Peptides-Func, Peptides-Struct, PCQM-Contact.

| Hyperparmeter | PascalVOC-SP | COCO-SP | Peptides-Func | Peptides-Struct | PCQM-Contact |
|---|---|---|---|---|---|
| #Layers | 6 | 7 | 4 | 4 | 7 |
| Hidden dim | 80 | 56 | 96 | 96 | 64 |
| # Heads | 1 | 2 | 4 | 4 | 4 |
| Dropout | 0.15 | 0.0 | 0.1 | 0.15 | 0.0 |
| Attention dropout | 0.5 | 0.5 | 0.1 | 0.5 | 0.56 |
| PE | LapPE | LapPE | RWSE | RWSE | LapPE |
| PE dim | 16 | 16 | 16 | 20 | 16 |
| Batch size | 200 | 150 | 200 | 200 | 128 |
| Learning rate | 1e-3 | 1e-3 | 1e-3 | 1e-3 | 3e-4 |
| #Epochs | 300 | 300 | 250 | 250 | 250 |
| Weight decay | 1e-5 | 1e-2 | 1e-2 | 1e-2 | 0.0 |
| $N_s$ | 30 | 20 | 30 | 40 | 30 |
| $\eta$ | 0.1 | 0.1 | 0.1 | 0.1 | 0.1 |
| $s$ | 30 | 20 | 30 | 40 | 30 |
| #Parameters | 508305 | 315305 | 470693 | 468783 | 386526 |

Table A3 Hyperparameters used in Primphormer for datasets: CIFAR10, MNIST, MalNet-Tiny, PATTERN, CLUSTER.

| Hyperparmeter | CIFAR10 | MNIST | MalNet-Tiny | PATTERN | CLUSTER |
|---|---|---|---|---|---|
| #Layers | 3 | 4 | 5 | 6 | 12 |
| Hidden dim | 52 | 40 | 84 | 48 | 52 |
| #Heads | 1 | 1 | 1 | 1 | 1 |
| Dropout | 0.15 | 0.1 | 0.15 | 0.0 | 0.15 |
| Attention dropout | 0.5 | 0.5 | 0.5 | 0.5 | 0.5 |
| PE | ESLapPE | ESLapPE | - | ESLapPE | ESLapPE |
| PE dim | 8 | 8 | - | 8 | 10 |
| Batch size | 200 | 200 | 64 | 128 | 48 |
| Learning rate | 1e-3 | 1e-3 | 1e-3 | 1e-3 | 1e-3 |
| #Epochs | 300 | 300 | 300 | 200 | 300 |
| Weight decay | 1e-2 | 1e-5 | 1e-3 | 1e-5 | 1e-5 |
| $N_s$ | 20 | 30 | 50 | 30 | 40 |
| $\eta$ | 0.1 | 0.1 | 0.1 | 0.1 | 0.1 |
| $s$ | 20 | 30 | 50 | 30 | 40 |
| #Parameters | 112957 | 101714 | 519605 | 208387 | 499386 |

Table A4 Hyperparameters used in Table. 4.

| Hyperparmeter | CIFAR10 | MalNet-Tiny | PasvalVOC-SP | Peptides-Func | ogbn-products |
|---|---|---|---|---|---|
| #Layers | 5 | 5 | 4 | 4 | 2 |
| Hidden dim | 40 | 64 | 96 | 96 | 128 |
| Batch size | 128 | 4 | 32 | 128 | - |

## C  PROOFS OF THEORETICAL RESULTS

In this section, we provide the proofs of theoretical results in this paper.

### C.1  PROOF DETAILS OF THEOREM 1

The Lagrangian of (2.5) is defined by,

$$\mathcal{L}(\boldsymbol{W}_e, \boldsymbol{W}_r, \boldsymbol{e}_i, \boldsymbol{r}_j, \boldsymbol{h}_{e_i}, \boldsymbol{h}_{r_j}) = \frac{1}{2} \sum_{i=1}^{N} \boldsymbol{e}_i^\top \boldsymbol{\Lambda} \boldsymbol{e}_i + \frac{1}{2} \sum_{j=1}^{N} \boldsymbol{r}_j^\top \boldsymbol{\Lambda} \boldsymbol{r}_j - \mathrm{Tr}(\boldsymbol{W}_e^\top \boldsymbol{W}_r)$$

$$- \sum_{i=1}^{N} \boldsymbol{h}_{e_i}^\top \big(\boldsymbol{e}_i - f_X \boldsymbol{W}_e \phi_q(\boldsymbol{x}_i)\big) - \boldsymbol{h}_{r_j}^\top \big(\boldsymbol{r}_j - f_X \boldsymbol{W}_r \phi_k(\boldsymbol{x}_j)\big),$$

(C1)

where $\boldsymbol{h}_{e_i}, \boldsymbol{h}_{r_j} \in \mathbb{R}^s s$ are dual variable vectors corresponding to the equality constraints regarding the projection scores $\boldsymbol{e}_i$ and $\boldsymbol{r}_j$.

By taking the partial derivatives to the Lagrangian (C1), the Karush-Kuhn-Tucker (KKT) conditions lead to the following equalities,

$$\begin{cases} \dfrac{\partial \mathcal{L}}{\partial \boldsymbol{W}_e} = 0 \Rightarrow \boldsymbol{W}_r = \sum_{i=1}^{N} f_X^\top \boldsymbol{h}_{e_i} \phi_q(\boldsymbol{x}_i)^\top \\[2mm] \dfrac{\partial \mathcal{L}}{\partial \boldsymbol{W}_r} = 0 \Rightarrow \boldsymbol{W}_e = \sum_{j=1}^{N} f_X^\top \boldsymbol{h}_{r_j} \phi_k(\boldsymbol{x}_j)^\top \\[2mm] \dfrac{\partial \mathcal{L}}{\partial \boldsymbol{e}_i} = 0 \Rightarrow \boldsymbol{\Lambda} \boldsymbol{e}_i = \boldsymbol{h}_{e_i}, \quad i \in [N] \\[2mm] \dfrac{\partial \mathcal{L}}{\partial \boldsymbol{r}_j} = 0 \Rightarrow \boldsymbol{\Lambda} \boldsymbol{r}_j = \boldsymbol{h}_{r_j}, \quad j \in [N] \\[2mm] \dfrac{\partial \mathcal{L}}{\partial \boldsymbol{h}_{e_i}} = 0 \Rightarrow \boldsymbol{e}_i = f_X \boldsymbol{W}_e \phi_q(\boldsymbol{x}_i), \quad i \in [N] \\[2mm] \dfrac{\partial \mathcal{L}}{\partial \boldsymbol{h}_{r_j}} = 0 \Rightarrow \boldsymbol{r}_j = f_X \boldsymbol{W}_r \phi_k(\boldsymbol{x}_j), \quad j \in [N]. \end{cases}$$

(C2)

By eliminating the primal variables $\boldsymbol{W}_e$ and $\boldsymbol{W}_r$, we have,

$$\begin{cases} \sum_{j=1}^{N} \boldsymbol{F}_X \boldsymbol{h}_{r_j} \phi_k(\boldsymbol{x}_j)^\top \phi_q(\boldsymbol{x}_i) = \boldsymbol{\Lambda}^{-1} \boldsymbol{h}_{e_i}, \quad i \in [N], \\[2mm] \sum_{i=1}^{N} \boldsymbol{F}_X \boldsymbol{h}_{e_i} \phi_q(x_i)^\top \phi_k(\boldsymbol{x}_j) = \boldsymbol{\Lambda}^{-1} \boldsymbol{h}_{r_j}, \quad j \in [N], \end{cases}$$

(C3)

where $\boldsymbol{F}_X := f_X f_X^\top \in \mathbb{S}_+^{s \times s}$ is the auto-correlation matrix. It can be expressed in the following matrix form,

$$\begin{bmatrix} \boldsymbol{0}_{N \times N} & [\phi_q(\boldsymbol{x}_i)^\top \phi_k(\boldsymbol{x}_j)] \\ [\phi_k(\boldsymbol{x}_j)^\top \phi_q(\boldsymbol{x}_i)] & \boldsymbol{0}_{N \times N} \end{bmatrix} \begin{bmatrix} \boldsymbol{H}_e \\ \boldsymbol{H}_r \end{bmatrix} \boldsymbol{F}_X = \begin{bmatrix} \boldsymbol{H}_e \\ \boldsymbol{H}_r \end{bmatrix} \boldsymbol{\Lambda}^{-1},$$

(C4)

with $\boldsymbol{H}_e := [\boldsymbol{h}_{e_1}, \ldots, \boldsymbol{h}_{e_N}]^\top \in \mathbb{R}^{N \times s}$, and $\boldsymbol{H}_r := [\boldsymbol{h}_{r_1}, \ldots, \boldsymbol{h}_{r_N}]^\top \in \mathbb{R}^{N \times s}$.

Then it can be noticed that the KSVD optimization problem in the dual space yields the following generalized eigenvalue problem with an asymmetric kernel $\boldsymbol{K}$,

$$\boldsymbol{K}\boldsymbol{H}_r\boldsymbol{F}_X = \boldsymbol{H}_e\boldsymbol{\Sigma},$$
$$\boldsymbol{K}^\top\boldsymbol{H}_e\boldsymbol{F}_X = \boldsymbol{H}_r\boldsymbol{\Sigma}, \tag{C5}$$

which collects the solutions corresponding to the non-zero entries in $\boldsymbol{\Lambda}$ such that $\boldsymbol{\Sigma} \triangleq \boldsymbol{\Lambda}^{-1}$. The asymmetric kernel matrix $\boldsymbol{K}$, induced by $\boldsymbol{K}_{ij} := \langle \phi_q(\boldsymbol{x}_i), \phi_k(\boldsymbol{x}_j) \rangle, \forall i, j \in [N]$, corresponds to the attention matrix.

## C.2 DERIVATION OF SCORES (2.8) IN THE PRIMAL AND DUAL SPACES

With the derivations and KKT conditions of the primal-dual optimization above, the primal and dual representation for the self-attention can be formulated as follows,

$$\text{Primal} : \begin{cases} \boldsymbol{e}(\boldsymbol{x}) = f_X\boldsymbol{W}_e\phi_q(\boldsymbol{x}), \\ \boldsymbol{r}(\boldsymbol{x}) = f_X\boldsymbol{W}_r\phi_k(\boldsymbol{x}). \end{cases} \tag{C6}$$

$$\text{Dual} : \begin{cases} \boldsymbol{e}(\boldsymbol{x}) = f_X\boldsymbol{W}_e\phi_q(\boldsymbol{x}_i) = \sum_{j=1}^{N} \boldsymbol{F}_X\boldsymbol{h}_{r_j}\phi_k(\boldsymbol{x}_j)^\top\phi_q(\boldsymbol{x}), \\ \boldsymbol{r}(\boldsymbol{x}) = f_X\boldsymbol{W}_r\phi_k(\boldsymbol{x}_i) = \sum_{i=1}^{N} \boldsymbol{F}_X\boldsymbol{h}_{e_i}\phi_q(\boldsymbol{x}_i)^\top\phi_k(\boldsymbol{x}). \end{cases} \tag{C7}$$

Then, the primal and dual representations for self-attention can be folumated as follows,

$$\text{Primal} : \begin{cases} \boldsymbol{e}(\boldsymbol{x}) = \boldsymbol{W}_{e|X}^\top\phi_q(\boldsymbol{x}), \\ \boldsymbol{r}(\boldsymbol{x}) = \boldsymbol{W}_{r|X}^\top\phi_k(\boldsymbol{x}), \end{cases} \quad \text{Dual} : \begin{cases} \boldsymbol{e}(\boldsymbol{x}) = \sum_{j=1}^{N} \tilde{\boldsymbol{h}}_{r_j}\kappa(\boldsymbol{x}, \boldsymbol{x}_j), \\ \boldsymbol{r}(\boldsymbol{x}) = \sum_{i=1}^{N} \tilde{\boldsymbol{h}}_{e_i}\kappa(\boldsymbol{x}_i, \boldsymbol{x}), \end{cases} \tag{C8}$$

where $\boldsymbol{W}_{e|X}^\top := f_X\boldsymbol{W}_e \in \mathbb{R}^{s\times p}$, $\boldsymbol{W}_{r|X}^\top := f_X\boldsymbol{W}_r \in \mathbb{R}^{s\times p}$ and $\tilde{\boldsymbol{h}}_{r_j} := \boldsymbol{F}_X\boldsymbol{h}_{r_j}, \tilde{\boldsymbol{h}}_{e_i} := \boldsymbol{F}_X\boldsymbol{h}_{e_i}$ are values for self-attention, respectively.

## C.3 PROOF DETAILS OF THEOREM 2

*Proof.* Based on the KKT conditions (C2) and (2.6), the objective on stationary points is,

$$\begin{aligned} J &= \frac{1}{2}\sum_{i=1}^{N} \boldsymbol{e}_i^\top\boldsymbol{\Lambda}\boldsymbol{e}_i + \frac{1}{2}\sum_{j=1}^{N} \boldsymbol{r}_j^\top\boldsymbol{\Lambda}\boldsymbol{r}_j - \text{Tr}\left(\boldsymbol{W}_e^\top\boldsymbol{W}_r\right) \\ &= \frac{1}{2}\sum_{i=1}^{N} \left(\boldsymbol{\Lambda}^{-1}\boldsymbol{h}_{e_i}\right)^\top\boldsymbol{\Lambda}\boldsymbol{\Lambda}^{-1}\boldsymbol{h}_{e_i} + \frac{1}{2}\sum_{j=1}^{N} \left(\boldsymbol{\Lambda}^{-1}\boldsymbol{h}_{r_j}\right)^\top\boldsymbol{\Lambda}\boldsymbol{\Lambda}^{-1}\boldsymbol{h}_{r_j} \\ &\quad - \text{Tr}\left(\left(\sum_{j=1}^{N} \phi_k(\boldsymbol{x}_j)\boldsymbol{h}_{r_j}^\top f_X\right) \cdot \left(\sum_{i=1}^{N} f_X^\top\boldsymbol{h}_{e_i}\phi_q(\boldsymbol{x}_i)^\top\right)\right) \\ &= \frac{1}{2}\sum_{i=1}^{N} \boldsymbol{h}_{e_i}^\top\boldsymbol{\Lambda}^{-1}\boldsymbol{h}_{e_i} + \frac{1}{2}\sum_{j=1}^{N} \boldsymbol{h}_{r_j}^\top\boldsymbol{\Lambda}^{-1}\boldsymbol{h}_{r_j} - \text{Tr}\left(\sum_{i,j} \phi_k(\boldsymbol{x}_j)\boldsymbol{h}_{r_j}^\top\boldsymbol{F}_X\boldsymbol{h}_{e_i}\phi_q(\boldsymbol{x}_i)^\top\right) \\ &= \frac{1}{2}\text{Tr}\left(\boldsymbol{H}_e\boldsymbol{\Sigma}\boldsymbol{H}_e^\top\right) + \frac{1}{2}\text{Tr}\left(\boldsymbol{H}_r\boldsymbol{\Sigma}\boldsymbol{H}_r^\top\right) - \text{Tr}\left(\sum_{i,j} \phi_q(\boldsymbol{x}_i)^\top\phi_k(\boldsymbol{x}_j)\boldsymbol{h}_{r_j}^\top\boldsymbol{F}_X\boldsymbol{h}_{e_i}\right) \\ &= \frac{1}{2}\text{Tr}\left(\boldsymbol{H}_e\boldsymbol{\Sigma}\boldsymbol{H}_e^\top\right) + \frac{1}{2}\text{Tr}\left(\boldsymbol{H}_r\boldsymbol{\Sigma}\boldsymbol{H}_r^\top\right) - \text{Tr}\left(\boldsymbol{K}\boldsymbol{H}_r\boldsymbol{F}_X\boldsymbol{H}_e^\top\right) \\ &= \frac{1}{2}\text{Tr}\left(\boldsymbol{K}\boldsymbol{H}_r\boldsymbol{F}_X\boldsymbol{H}_e^\top\right) + \frac{1}{2}\text{Tr}\left(\boldsymbol{K}^\top\boldsymbol{H}_e\boldsymbol{F}_X\boldsymbol{H}_r^\top\right) - \text{Tr}\left(\boldsymbol{K}\boldsymbol{H}_r\boldsymbol{F}_X\boldsymbol{H}_e^\top\right) \\ &= \frac{1}{2}\text{Tr}\left(\boldsymbol{K}^\top\boldsymbol{H}_e\boldsymbol{F}_X\boldsymbol{H}_r^\top\right) - \frac{1}{2}\text{Tr}\left(\boldsymbol{K}\boldsymbol{H}_r\boldsymbol{F}_X\boldsymbol{H}_e^\top\right) \\ &= \frac{1}{2}\text{Tr}\left(\boldsymbol{H}_e\boldsymbol{F}_X\boldsymbol{H}_r^\top\boldsymbol{K}^\top\right) - \frac{1}{2}\text{Tr}\left(\boldsymbol{K}\boldsymbol{H}_r\boldsymbol{F}_X\boldsymbol{H}_e^\top\right) \\ &= \frac{1}{2}\text{Tr}\left(\left(\boldsymbol{H}_e\boldsymbol{F}_X\boldsymbol{H}_r^\top\boldsymbol{K}^\top\right)^\top\right) - \frac{1}{2}\text{Tr}\left(\boldsymbol{K}\boldsymbol{H}_r\boldsymbol{F}_X\boldsymbol{H}_e^\top\right) \\ &= \frac{1}{2}\text{Tr}\left(\boldsymbol{K}\boldsymbol{H}_r\boldsymbol{F}_X\boldsymbol{H}_e^\top\right) - \frac{1}{2}\text{Tr}\left(\boldsymbol{K}\boldsymbol{H}_r\boldsymbol{F}_X\boldsymbol{H}_e^\top\right) = 0. \end{aligned}$$

$$\tag{C9}$$

This completes the proof. $\square$

## C.4 PROOF DETAILS OF THEOREM 3

*Proof.* The proof follows ideas in (Alberti et al., 2023). We first introduce the Sumformer $\mathcal{S}$ and we divide the approximation into two parts: 1) approximate $f$ by a $\mathcal{S}$ and 2) approximate $\mathcal{S}$ by a Primphormer $\mathcal{T}_{\mathrm{Pri}}$.

**Definition 3** (Sumformer). *Let $d' \in \mathbb{N}$ and let there be two functions $\phi : \mathcal{X} \to \mathbb{R}^{d'}$, $\psi : \mathcal{X} \times \mathbb{R}^{d'} \to \mathcal{Y}$. A Sumformer is a sequence-to-sequence function $\mathcal{S} : \mathcal{X}^N \to \mathcal{Y}^N$ which is evaluated by first computing*

$$\Xi := \sum_{k=1}^{N} \boldsymbol{\xi}(\boldsymbol{x}_k), \tag{C10}$$

*and then*

$$\mathcal{S}([\boldsymbol{x}_1, \cdots, \boldsymbol{x}_N]) := [\psi(\boldsymbol{x}_1, \Xi), \cdots, \psi(\boldsymbol{x}_N, \Xi)]. \tag{C11}$$

**Theorem 5** (Universal approximation of Sumformer). *For each function $f \in \mathcal{F}_{\mathrm{eq}}^N(\mathcal{X}, \mathcal{Y})$ and for each $\epsilon > 0$ there exists a Sumformer $\mathcal{S}$ such that*

$$\sup_{\boldsymbol{X} \in \mathcal{X}^N} \|f(\boldsymbol{X}) - \mathcal{S}(\boldsymbol{X})\|_\infty < \epsilon. \tag{C12}$$

We divide the approximation into two steps by the triangular inequality: 1) approximate $f$ by a Sumformer $\mathcal{S}$ and 2) approximate $\mathcal{S}$ by a Primphormer $\mathcal{T}_{\mathrm{Pri}}$.

$$\sup_{\boldsymbol{X} \in \mathcal{X}^N} \|f(\boldsymbol{X}) - \mathcal{T}_{\mathrm{Pri}}(\boldsymbol{X})\|_\infty \leq \sup_{\boldsymbol{X} \in \mathcal{X}^N} \|f(\boldsymbol{X}) - \mathcal{S}(\boldsymbol{X})\|_\infty + \sup_{\boldsymbol{X} \in \mathcal{X}^N} \|\mathcal{S}(\boldsymbol{X}) - \mathcal{T}_{\mathrm{Pri}}(\boldsymbol{X})\|_\infty. \tag{C13}$$

According to Theorem 5, we know that there exists a Sumformer $\mathcal{S}$ which approximates $f$ to an error of $\epsilon/2$. This Sumformer has the inherent latent dimension $d'$.

Secondly, we turn to the second term and construct a Primphormer that is able to approximate Sumformer to $\epsilon/2$ error. The structure of Transformer is $\boldsymbol{X} + \mathrm{FC}\,(\boldsymbol{X} + \mathrm{Att}(\boldsymbol{X}))$ where FC and Att are the fully-connected and self-attention modules, respectively. The attention map $\mathrm{Att}(\boldsymbol{X})$ of Primphormer is calculated in the primal space (2.8) and the rest of the architecture in Primphormer stays the same. Here, we follow the proof idea proposed in (Alberti et al., 2023) and refer readers to this work for detailed information on the theoretical result.

We have the input $\boldsymbol{X} = [\boldsymbol{x}_1, \cdots, \boldsymbol{x}_N] \in \mathcal{X}^N$ with $\boldsymbol{x}_i \in \mathbb{R}^d$. Set the attention in the first layers to zero, we obtain the feed-forward layers without attention. We first map $\boldsymbol{X}$ with a feed-forward transformation to

$$\begin{bmatrix} \boldsymbol{x}_1 & \cdots & \boldsymbol{x}_N \\ \boldsymbol{x}_1 & \cdots & \boldsymbol{x}_N \end{bmatrix} \in \mathbb{R}^{2d \times N}. \tag{C14}$$

Then, a two-layer feed-forward network can be constructed to act as the identity on the first $N$ components while approximating the function $\boldsymbol{\xi}$ in Sumformer (Hornik et al., 1989; Alberti et al., 2023). We have.

$$\begin{bmatrix} \boldsymbol{x}_1 & \cdots & \boldsymbol{x}_N \\ \boldsymbol{\xi}(\boldsymbol{x}_1) & \cdots & \boldsymbol{\xi}(\boldsymbol{x}_N) \end{bmatrix} \in \mathbb{R}^{(d+d') \times N}. \tag{C15}$$

Before getting to the second step, we we add a linear mapping with

$$\begin{cases} \boldsymbol{W} = \begin{bmatrix} \boldsymbol{0}_{d \times 1} & \boldsymbol{I}_d & \boldsymbol{0}_{d \times d'} & \boldsymbol{0}_{d \times d'} \\ \boldsymbol{0}_{d' \times 1} & \boldsymbol{0}_{d' \times d} & \boldsymbol{I}_{d'} & \boldsymbol{0}_{d' \times d'} \end{bmatrix}^\top \in \mathbb{R}^{(1+d+2d') \times (d+d')}, \\ \boldsymbol{b} = \begin{bmatrix} \boldsymbol{1}_N & \boldsymbol{0}_{N \times (d+2d')} \end{bmatrix}^\top \in \mathbb{R}^{(1+d+2d') \times N}, \end{cases} \tag{C16}$$

and get an output after the first step:

$$\begin{bmatrix} 1 & \cdots & 1 \\ \boldsymbol{x}_1 & \cdots & \boldsymbol{x}_N \\ \boldsymbol{\xi}(\boldsymbol{x}_1) & \cdots & \boldsymbol{\xi}(\boldsymbol{x}_N) \\ \boldsymbol{0}_{d' \times 1} & \cdots & \boldsymbol{0}_{d' \times 1} \end{bmatrix} \in \mathbb{R}^{(1+d+2d') \times N}. \tag{C17}$$

Secondly, we turn to attention scheme to represent the sum $\boldsymbol{\Xi} = \sum_{i=1}^{N} \boldsymbol{\xi}(\boldsymbol{x}_i)$ defined in the definition (3). Set $\boldsymbol{W}_q = \boldsymbol{W}_k = [\boldsymbol{e}_1, \boldsymbol{0}_{(1+d+2d') \times (d+2d')}]$ with $\boldsymbol{e}_1 = [1, \boldsymbol{0}_{1 \times (d+2d')}]^\top$. we have,

$$\phi_q(\boldsymbol{X}_1) = \phi_k(\boldsymbol{X}_1) = \left[\boldsymbol{1}_{N \times 1}, \boldsymbol{0}_{N \times (d+2d')}\right]^\top \in \mathbb{R}^{(1+d+2d') \times N}. \tag{C18}$$

Let the data-dependent projection $f(\boldsymbol{X}) = \boldsymbol{B} \boldsymbol{X} \boldsymbol{1}_N \boldsymbol{1}_{N_s}^\top$ with $\boldsymbol{B} = [\boldsymbol{0}_{d' \times 1}, \boldsymbol{0}_{d' \times d}, \boldsymbol{I}_{d'}, \boldsymbol{0}_{d' \times d'}]$, we have,

$$f(\boldsymbol{X}) = \overbrace{\left[\sum_{i=1}^{N} \boldsymbol{\xi}(\boldsymbol{x}_i), \cdots, \sum_{i=1}^{N} \boldsymbol{\xi}(\boldsymbol{x}_i)\right]}^{N_s} = [\boldsymbol{\Xi}, \cdots, \boldsymbol{\Xi}] \in \mathbb{R}^{d' \times N_s}. \tag{C19}$$

Let $\boldsymbol{W}_e = \boldsymbol{W}_r = [\boldsymbol{e}_1, \boldsymbol{0}_{(1+d+2d') \times (N_s-1)}]^\top$, the projection scores in (2.8) are

$$\begin{cases} \boldsymbol{e}(\boldsymbol{X}_1) = f(\boldsymbol{X}_1)\boldsymbol{W}_e \phi_q(\boldsymbol{X}_1) & = [\boldsymbol{\Xi}, \cdots, \boldsymbol{\Xi}] \in \mathbb{R}^{d' \times N}. \\ \boldsymbol{r}(\boldsymbol{X}_1) = f(\boldsymbol{X}_1)\boldsymbol{W}_r \phi_k(\boldsymbol{X}_1) & = [\boldsymbol{\Xi}, \cdots, \boldsymbol{\Xi}] \in \mathbb{R}^{d' \times N}. \end{cases} \tag{C20}$$

To fit the dimension of the output, we concatenate the projection scores $[\boldsymbol{e}(\boldsymbol{X}_1); \boldsymbol{r}(\boldsymbol{X}_1)] \in \mathbb{R}^{2d' \times N}$, and choose a compatibility matrix $\boldsymbol{W}_c = [\boldsymbol{0}_{(1+d+d') \times 2d'}; \frac{1}{2}\boldsymbol{I}_{d'}, \frac{1}{2}\boldsymbol{I}_{d'}] \in \mathbb{R}^{(1+d+2d') \times 2d'}$, such that

$$\boldsymbol{o}(\boldsymbol{X}_1) = \boldsymbol{W}_c \begin{bmatrix} \boldsymbol{e}(\boldsymbol{X}_1) \\ \boldsymbol{r}(\boldsymbol{X}_1) \end{bmatrix} = \begin{bmatrix} \boldsymbol{0}_{(1+d+d') \times 1} & \cdots & \boldsymbol{0}_{(1+d+d') \times 1} \\ \boldsymbol{\Xi} & \cdots & \boldsymbol{\Xi} \end{bmatrix} \in \mathbb{R}^{(1+d+2d') \times N}. \tag{C21}$$

Then apply a residual connection and obtain the same output as outlined in (Alberti et al., 2023),

$$\begin{bmatrix} 1 & \cdots & 1 \\ \boldsymbol{x}_1 & \cdots & \boldsymbol{x}_N \\ \boldsymbol{\xi}(\boldsymbol{x}_1) & \cdots & \boldsymbol{\xi}(\boldsymbol{x}_N) \\ \boldsymbol{\Xi} & \cdots & \boldsymbol{\Xi} \end{bmatrix} \in \mathbb{R}^{(1+d+2d') \times N}. \tag{C22}$$

Because only the attention map $\mathrm{Att}(\mathrm{X})$ is changed in the architecture and the rest stays the same, the construction of $\psi$ is as same as that in (Alberti et al., 2023), i.e., $\mathcal{O}(N(\frac{1}{\epsilon})^{dN}/N!)$ feed-forward layers for approximating $\psi$ in the discontinuous case and two feed-forward layers for approximating $\psi$ in the continuous case. Above all, we can construct a Primphormer that approximates the Sumformer to $\epsilon/2$ error. $\qquad \square$

## C.5 Proof details of Theorem 4

*Proof.* The proof can be done in a similar way as Theorem 3. Firstly, let the target function $f(\boldsymbol{X}) := [g(\boldsymbol{x}_1, \{\boldsymbol{x}_2, \cdots, \boldsymbol{x}_N\}), \cdots, g(\boldsymbol{x}_N, \{\boldsymbol{x}_1, \cdots, \boldsymbol{x}_{N-1}\})]$. Since the target function $f$ is continuous, its component functions $f_1, \cdots, f_N$, i.e., $g$, are also continuous. The compactness of $\mathcal{X}$ shows that $\mathcal{X}^N$ is also compact and therefore $g$ is uniformly continuous. Without loss of generality, let the compact support of $g$ be contained in $[0, 1]^{d \times N}$. Then we can define a piece-wise constant function $\overline{g}$ by

$$\overline{g}(\boldsymbol{X}) = \sum_{\boldsymbol{P} \in \mathbb{G}_\delta} g(\boldsymbol{P}) \boldsymbol{1}\{\boldsymbol{X} \in C_{\boldsymbol{P}}\}, \tag{C23}$$

where the grid $\mathbb{G}_\delta := \{0, \delta, \cdots, 1 - \delta\}^{d \times N}$ for some $\delta := \frac{1}{\Delta}$ with $\Delta \in \mathbb{N}$ consisting of cubes $C_{\boldsymbol{P}} = \prod_{i=1}^{N} \prod_{k=1}^{d} [\boldsymbol{P}_{i,k}, \boldsymbol{P}_{i,k} + \delta)$. Because $g$ is uniformly continuous, for each $\epsilon > 0$, there exists a $\delta > 0$ such that

$$\sup_{\boldsymbol{X} \in \mathcal{X}^N} \|g(\boldsymbol{X}) - \overline{g}(\boldsymbol{X})\|_\infty < \epsilon. \tag{C24}$$

Secondly, choose the positional encoding

$$\boldsymbol{E} = \begin{bmatrix} 0 & 1 & 2 & \cdots & N-1 \\ 0 & 1 & 2 & \cdots & N-1 \\ \vdots & \vdots & \vdots & & \vdots \\ 0 & 1 & 2 & \cdots & N-1 \end{bmatrix} \in \mathbb{R}^{d \times N}. \tag{C25}$$

After applying the quantization, the output is in the following set,

$$\mathbb{H}_\delta := \left\{\boldsymbol{P} + \boldsymbol{E} \in \mathbb{R}^{d \times N} | \boldsymbol{P} \in \mathbb{G}_\delta\right\}. \tag{C26}$$

Then the $i$-th column of $\boldsymbol{X} + \boldsymbol{E}$ is in the range $[i-1, i)^d$, meaning that the entries corresponding to different tokens lie in disjoint intervals. More precisely, for any $\boldsymbol{H} \in \mathbb{G}_\delta$, its $i$-th column $\boldsymbol{H}_i \in [i-1 : \delta : i - \delta]$.

Consider a vector $\boldsymbol{u} = \frac{1-\delta}{N\delta^{-d+1}} \times \left(1, \delta^{-1}, \cdots, \delta^{-d+1}\right) \in \mathbb{R}^d$. It is easy to check that for any $\boldsymbol{H} \in \mathbb{G}_\delta$, the map $l(\boldsymbol{H}_i) = \boldsymbol{u}^\top \boldsymbol{H}_i$ is one-to-one,

$$\boldsymbol{u}^\top \boldsymbol{H}_i \in \left[ \frac{(1-\delta)(i-1)}{N\delta^{-d+1}} \sum_{k=0}^{d-1} \delta^{-k} : \frac{(1-\delta)}{N\delta^{-d}} : \frac{(1-\delta)i}{N\delta^{-d+1}} \sum_{k=0}^{d-1} \delta^{-k} - \frac{(\delta^{-d}-1)}{N\delta^{-d-1}} \right]. \tag{C27}$$

Therefore, for each column $\boldsymbol{H}_i$, the image of $l(\boldsymbol{H}_i)$ is in an interval disjoint from the other columns. We can know that $l(\boldsymbol{H}_i)$ can be thought as a "column id" for different columns, for any permutation $\pi : [N] \to [N]$,

$$l\left(\boldsymbol{H}_{\pi(1)}\right) < l\left(\boldsymbol{H}_{\pi(2)}\right) < \cdots < l\left(\boldsymbol{H}_{\pi(N)}\right). \tag{C28}$$

Besides, it can be easily checked that the image of $l$ lies within the interval $[0, 1]$,

$$0 \le l\left(\boldsymbol{H}_{\pi(1)}\right) < l\left(\boldsymbol{H}_{\pi(2)}\right) < \cdots < l\left(\boldsymbol{H}_{\pi(N)}\right) < 1. \tag{C29}$$

Next, we want to represent $\overline{g}$ using an appropriate $\mathcal{S}$. Without loss of generality, we choose the $k$-th component of $f$, i.e., $\overline{g}(\boldsymbol{x}_k, \{\boldsymbol{x}_i | i \ne k, i \in [N]\})$. Assign each grid point $\boldsymbol{H}$ a coordinate $\chi(\boldsymbol{H}) = \boldsymbol{b} \in [0, 1]^N$ by the construction of the function $l$. Let $\boldsymbol{b} = [l(\boldsymbol{H}_i) | i \in [N]] \in [0, 1]^N$. The map $\chi$ is bijective and there are finitely many $\boldsymbol{b}$. We can enumerate all $\boldsymbol{b}$ using a function $\mu : [0, 1]^N \to \mathbb{N}$. This function could be represented by the Kolmogorov-Arnold representation theorem (Khesin & Tabachnikov, 2014; Zaheer et al., 2017), as stated below,

**Theorem 6** (Kolmogorov-Arnold representation)**.** *Let $f : [0, 1]^N \to \mathbb{R}$ be an arbitrary multivariate continuous function iff it has the representation,*

$$f(\boldsymbol{x}_1, \cdots, \boldsymbol{x}_N) = \rho\left(\sum_{n=1}^{N} \lambda_n \phi(\boldsymbol{x}_n)\right) \tag{C30}$$

*with continuous outer and inner functions $\rho : \mathbb{R}^{2N+1} \to \mathbb{R}$ and $\phi : \mathbb{R} \to \mathbb{R}^{2N+1}$. The inner function $\phi$ is independent of the function $f$.*

Now, we can utilize Theorem 6 to find the representation for the function $\mu$,

$$\mu(\boldsymbol{b}) = \rho\left(\sum_{n=1}^{N} \lambda_n \phi(\boldsymbol{b}_n)\right). \tag{C31}$$

Define $\boldsymbol{\Xi} := \sum_{n=1}^{N} \boldsymbol{\xi}(\boldsymbol{b}_n) = \sum_{n=1}^{N} \lambda_n \phi(\boldsymbol{b}_n)$ and a quantization function $q$ such that $\boldsymbol{b}_n = l(q(\boldsymbol{x}_n + \boldsymbol{E}_n))$. It is feasible because $\boldsymbol{b}_n$ varies for different indices, as claimed in "column id" (C28). Now we can recover the grid $\boldsymbol{H}$,

$$\boldsymbol{H} = \chi^{-1} \circ \mu^{-1} \circ \rho(\boldsymbol{\Xi}). \tag{C32}$$

We then define the function $\psi$ such that the related $\mathcal{S}$ is equal to $\overline{g}$:

$$\psi(\boldsymbol{x}_k, \boldsymbol{\Xi}) := \overline{g}\left(\iota(\chi^{-1} \circ \mu^{-1} \circ \rho(\boldsymbol{\Xi}) - \boldsymbol{E})\right), \tag{C33}$$

with $\iota : \boldsymbol{P} \mapsto (\boldsymbol{P}_k, \boldsymbol{P}_{i \ne k})$ to fit the input requirement of $\overline{g}$. Since we chose $\overline{g}$ to uniformly approximate $g$, i.e., each component of $f$ up to $\epsilon$ error, it implies that $\mathcal{S}$ with a positional encoding uniformly approximates $f$ up to $\epsilon$ error.

Thirdly, we need to prove the universal approximation between a Sumformer and a Primphormer after adding a positional encoding. The proof (C.4) still holds because it only involves the architecture. We can claim that there exists a Primphormer with a positional encoding $\mathcal{T}_{\text{PE}}$ uniformly approximating a Sumformer $\mathcal{S}$.

Above all, we end the proof by using the triangular inequality,

$$\sup_{\boldsymbol{X} \in \mathcal{X}^N} \|f(\boldsymbol{X}) - \mathcal{T}_{\text{PE}}(\boldsymbol{X})\|_\infty \le \sup_{\boldsymbol{X} \in \mathcal{X}^N} \|f(\boldsymbol{X}) - \mathcal{S}(\boldsymbol{X})\|_\infty + \sup_{\boldsymbol{X} \in \mathcal{X}^N} \|\mathcal{S}(\boldsymbol{X}) - \mathcal{T}_{\text{PE}}(\boldsymbol{X})\|_\infty < \epsilon. \tag{C34}$$

$\square$

# D PSEUDO-CODE

**Algorithm 1** PyTorch-like Pseudo-Code for Primphormer Module.

```python
import torch
import torch.nn as nn
import torch.nn.functional as F
from torch_geometric.nn import global_mean_pool
from torch_geometric.utils import to_dense_batch

class Primphormer(nn.Module):
    def __init__(self, in_dim, out_dim, n_heads, Ns, low_rank):
        super().__init__()
        self.d_keys = out_dim // n_heads # key dimension.
        self.q_proj = nn.Linear(in_dim, out_dim) # query
        self.k_proj = nn.Linear(in_dim, out_dim) # key
        self.vn_proj = nn.Linear(in_dim, out_dim) # virtual node
        self.n_heads = n_heads

        self.We = nn.Parameter(nn.init.orthogonal_(torch.Tensor(Ns, n_heads, self.d_keys)))
        self.Wr = nn.Parameter(nn.init.orthogonal_(torch.Tensor(Ns, n_heads, self.d_keys)))
        self.Lambda = nn.Parameter(nn.init.uniform_(torch.Tensor(n_heads, low_rank)))
        self.concate_weight = nn.Linear(2*low_rank, self.d_keys)

    def feature_map(self, Q, K):
        Q = F.normalized(Q, p=2, dim=-1)
        K = F.normalized(K, p=2, dim=-1)
        return Q, K

    def propagate_vn(self, batch, h):
        h = self.vn_proj(h)
        h_vn = global_mean_pool(h, batch.batch).unsqueeze(1) # aggregate by the virtual node.
        fx = h_vn + batch.fx # update f_X by the virtual node.
        return fx

    def forward(self, batch):
        x = batch.x
        x_dense, mask = to_dense_batch(x, batch.batch)
        B, M = mask.shape # batch, maximal #nodes
        fx = self.propagate_vn(batch, x)
        Q = self.q_proj(x_dense).view(B, M, self.n_heads, -1)
        K = self.k_proj(x_dense).view(B, M, self.n_heads, -1)
        Q, K = self.feature_map(Q, K)

        # compute data-dependent projections
        We_X = torch.einsum('bdv,vhe->bdhe', fx.transpose(2, 1), self.We)
        Wr_X = torch.einsum('bdv,vhe->bdhe', fx.transpose(2, 1), self.Wr)

        # compute projection scores
        escore = torch.einsum('bmhd,bhde->bmhe', Q, We_X.permute(0, 2, 3, 1))[mask]
        rscore = torch.einsum('bmhd,bhde->bmhe', K, Wr_X.permute(0, 2, 3, 1))[mask]

        score = torch.cat((escore, rscore), dim=-1)
        out = self.concate_weight(score).contiguous()
        out = out.view(-1, self.n_heads * self.d_keys) # final output
        batch.fx = fx #update for the next layer

        loss_escore = (torch.einsum('nhd,hd->nhd', escore,
            self.Lambda).norm(dim=-1,p=2)**2).mean() / 2
        loss_rscore = (torch.einsum('nhd,hd->nhd', rscore,
            self.Lambda).norm(dim=-1,p=2)**2).mean() / 2
        loss_trace = torch.einsum('dhe,ehk->dhk', self.We.permute(2, 1, 0),
            self.Wr).mean(dim=1).trace()
        loss_svd = (loss_escore + loss_rscore - loss_trace) ** 2

        return out, loss_svd
```

---

**Algorithm 2** Algorithm for Primphormer in the GPS architecture.

---

**Input:** Graph $G = (V, E)$ with $N$ nodes and $M$ edges; Adjacency matrix $\boldsymbol{A} \in \mathbb{R}^{N \times N}$; Node features $\boldsymbol{X} \in \mathbb{R}^{d_{\mathrm{n}} \times N}$, Edge features $\boldsymbol{E} \in \mathbb{R}^{d_{\mathrm{e}} \times M}$; Node and edge encoders; Local message passing model instance $\mathtt{MPNN}_e$; Primphormer model instance $\mathtt{Prim}$; Positional encoding function $f_{\mathrm{PE}}$; Layers $l \in [L - 1]$.

**Output:** Node representations $\boldsymbol{X}^L \in \mathbb{R}^{d \times N}$ and edge representations $\boldsymbol{E}^L \in \mathbb{R}^{d \times M}$ for downstream tasks.

1: $\boldsymbol{P}_{\mathrm{node}}, \boldsymbol{P}_{\mathrm{edge}} \leftarrow \varnothing$;
2: $\boldsymbol{P}_{\mathrm{node}}, \boldsymbol{P}_{\mathrm{edge}} \leftarrow f_{\mathrm{PE}}(\boldsymbol{X}, \boldsymbol{E})$
3: $\boldsymbol{X}^1 \leftarrow \bigoplus_{\mathrm{node}} (\mathtt{NodeEncoder}(\boldsymbol{X}), \boldsymbol{P}_{\mathrm{node}})$
4: $\boldsymbol{E}^1 \leftarrow \bigoplus_{\mathrm{edge}} (\mathtt{EdgeEncoder}(\boldsymbol{E}), \boldsymbol{P}_{\mathrm{edge}})$
5: **for** $l = 1, \cdots, L - 1$ **do**
6: $\quad \hat{\boldsymbol{X}}_M^{l+1}, \boldsymbol{E}^{l+1} \leftarrow \mathtt{MPNN}_e^l (\boldsymbol{X}^l, \boldsymbol{E}^l, \boldsymbol{A})$
7: $\quad \hat{\boldsymbol{X}}_P^{l+1} \leftarrow \mathtt{Prim}^l (\boldsymbol{X}^l)$
8: $\quad \boldsymbol{X}_M^{l+1} \leftarrow \mathtt{BatchNorm} \left( \mathtt{Dropout} \left( \hat{\boldsymbol{X}}_M^{l+1} \right) + \boldsymbol{X}^l \right)$
9: $\quad \boldsymbol{X}_P^{l+1} \leftarrow \mathtt{BatchNorm} \left( \mathtt{Dropout} \left( \hat{\boldsymbol{X}}_P^{l+1} \right) + \boldsymbol{X}^l \right)$
10: $\quad \boldsymbol{X}^{l+1} \leftarrow \mathtt{MLP}^l \left( \boldsymbol{X}_M^{l+1} + \boldsymbol{X}_P^{l+1} \right)$
11: **end for**
12: **return** $\boldsymbol{X}^L$ and $\boldsymbol{E}^L$

---

# E  ADDITIONAL EXPERIMENTS

We also conduct experiments to compare against more models (Ma et al., 2023; Tönshoff et al., 2023). Notably, Tönshoff et al. (2023) introduced an additional data preprocessing step (feature normalization, FN), which is parallel to our method and can be implemented similarly. We report the experimental results in Tables A5 and A6.

Table A5 Comparisons between our method and GRIT(Ma et al., 2023).

| Model | CIFAR10 | | | MNIST | | |
|---|---|---|---|---|---|---|
| GPS | ACC↑ | Time(s/epoch) | Memory(GB) | ACC↑ | Time(s/epoch) | Memory(GB) |
| Primphormer | $74.13 \pm 0.241$ | 32.6 | 2.74 | $98.56 \pm 0.042$ | 43.7 | 1.71 |
| GRIT(Ma et al., 2023) | $76.46 \pm 0.881$ | 158.8 | 22.8 | $98.11 \pm 0.111$ | 70.1 | 7.69 |

Table A6 Comparisons w/o FN between our method and GPS(Tönshoff et al., 2023).

| F1↑ | Ours | GPS | Ours+FN | GPS+FN |
|---|---|---|---|---|
| Pascal-VOC | $0.3980 \pm 0.0075$ | $0.3748 \pm 0.0109$ | $0.4602 \pm 0.0077$ | $0.4440 \pm 0.0065$ |
| COCO-SP | $0.3438 \pm 0.0046$ | $0.3412 \pm 0.0044$ | $0.3903 \pm 0.0061$ | $0.3884 \pm 0.0055$ |

