# OpenReview forum: "Primphormer: Leveraging Primal Representation for Graph Transformers"
_ICLR.cc/2025/Conference — Submitted to ICLR 2025_

### Official Review · Reviewer_6QH5 · 2024-11-01

**Soundness:** 3
**Presentation:** 3
**Contribution:** 3
**Rating:** 5
**Confidence:** 3

**Summary:**

The authors introduced Primphormer, a primal representation for graph transformers that eliminates the need for intensive pairwise computations by utilizing a kernel trick. This proposed technique has been demonstrated to serve as a universal approximator within a compact domain, showcasing superior performance compared to the current state-of-the-art.

**Strengths:**

* The authors introduce a novel primal representation for graph transformers, offering a comprehensive formulation that clearly delineates the distinctions between their method and traditional self-attention, which according to the paper relies on pairwise computations.
* The paper includes rigorous theoretical analysis and proofs that highlight the advantages of the proposed method, establishing its capability as a universal approximator.
* Extensive experiments were conducted, with results compared against benchmark models, demonstrating the significant performance improvements achieved by the proposed method.

**Weaknesses:**

* A minor concern arises regarding the notations used throughout the paper. A central explanation or summary may enhance reader comprehension, as there are instances where notations are utilized before being defined, or are left inadequately defined. For example, the notation ( N_s ) is introduced in the complexity analysis without prior definition.
* A fundamental issue regarding claims of computational complexity savings is the authors' assumption that all pairwise attentions in standard self-attention must be computed, which reflects an upper bound as indicated by big-O notation. In practice, attention mechanisms may focus only on local subgraphs or PPR sampled neighborhood, suggesting that neglecting very long-hop attention could have minimal impact. Consequently, the actual necessary computations may be significantly less than the proposed upper bound. It remains unclear whether this approximation or relax is applicable to the kernel trick mentioned.
* The significance of the universal approximation property is not adequately demonstrated in the paper and lacks experimental validation.
* In Figure 1(a), the necessity of residual connections prior to the merging of MPNN and ATTN is not well justified, raising concerns about the potential for added computational cost compared to applying the residual connections after the merge.

**Questions:**

See the detailed comments in the weakness part

---

> ### Author Response · Authors · 2024-11-20
>
> Thank you for your appreciation of the novelty of our primal representation for graph transformers. We also appreciate your recognition of the extensive experiments conducted in our work. We address your concerns as below:
>
> **R4.1 Clarity.**
>
> Thank you again for pointing out this issue. To address this, we will include clear definitions of the used variables in the Notations part at the beginning of Section 2.
>
> **R4.2 Discussion about long-range dependencies.**
>
> We agree that long-range interactions (LRI) may have minimal impact in certain scenarios. Nevertheless, there is a growing interest in tasks that necessitate LRI for optimal performance, where long-hop attention should not be neglected. This interest has led the development of the Long Range Graph Benchmark dataset [1], on which we also conduct experiments in the manuscript.
>
> Regarding local subgraphs or PPR sampled neighborhood schemes, we can adjust the formulation of the data-dependent projection $f\_X$ to accommodate them without disrupting the primal-dual relationship. For instance, we could enhance $f\_X$ to adapt to each node $i$, allowing $f\_{X,i}$ to aggregate local information. We believe this is a promising and interesting avenue for future research.
>
> **R4.3 Universal approximation property.**
>
> We sincerely appreciate your comment. The universal approximation property is a fundamental theoretical concept in deep learning theory. It is widely recognized that models that exhibit this property potentially possess strong generalization capabilities to unseen data or tasks. As we developed our Primphormer in the primal space, we were eager to determine whether our Primphormer still retains this advantageous characteristic. To explore this question, we introduced Theorems 3 and 4, offering theoretical assurances regarding the approximation abilities of our approach. This discussion will be included in the final revision. While directly demonstrating this property through experiments poses challenges, we hope the good performance of Primphormer presented in our manuscript's experiments could provide indirect evidence of this property.
>
> **R4.4 Residual connections.**
>
> In our manuscript, we maintain the same model architecture and residual connection scheme from [2, 3], substituting solely the attention module with our primal representation to ensure a fair comparison. We hope this clarification could address your concern.
>
>
> [1] Dwivedi V P, Rampášek L, Galkin M, et al. Long range graph benchmark[C]. NeurIPS, 2022.
>
> [2] Rampášek L, Galkin M, Dwivedi V P, et al. Recipe for a general, powerful, scalable graph transformer[C], NeurIPS, 2022.
>
> [3] Shirzad H, Velingker A, Venkatachalam B, et al. Exphormer: Sparse transformers for graphs[C], ICML, 2023.

---

> > ### Comment · Reviewer_6QH5 · 2024-11-25
> >
> > I would like to thank the authors for their well-organized responses to the comments. Your clarifications and contextual explanations are instrumental in re-evaluating the work, particularly concerning the identified gaps in the theoretical analysis and the experimental benefits. The integration of LRI alongside applied research practices enhances the meaningfulness of some analyses and aids in assessing the overall contributions of the proposed work. Explicitly outline the theoretical merits/gaps identified in the analysis and discuss how they could be filled in future work for experimental evaluation will strengthen the theoretical foundation of your research.

---

> > > ### Author Response · Authors · 2024-11-25
> > >
> > > Thank you very much for acknowledging our responses and engaging in insightful discussions.
> > >
> > > As you rightly pointed out, different tasks may entail varying requirements for interactions. Long-range interactions (LRI) may have minimal impact in certain scenarios where tasks involve only information exchange among nodes in the local neighborhood. However, in tasks such as the LRGB dataset, LRI may be either desired or necessary for learning tasks on graphs. We have integrated this discussion into the manuscript to enrich the depth of our analyses (Page 1, lines 27-32; Page 9, lines 466-468).
> > >
> > > We agree with your perspective on the importance of outlining the theoretical strengths and gaps to strengthen our research. Following your suggestion, we have explicitly outlined these aspects in our manuscript and also discussed how to address them in future work (Page 9, lines 454-456; 463-466).

---

### Official Review · Reviewer_9Dcv · 2024-11-03

**Soundness:** 3
**Presentation:** 2
**Contribution:** 3
**Rating:** 6
**Confidence:** 4

**Summary:**

This work introduces the use of primal representation for Graph Transformers, aiming to enhance computational efficiency. Inspired by a similar approach applied to sequences, the authors present a method tailored to graphs. They formulate the dual representation and explore the relationship between primal and dual forms. A theoretical analysis of the universal approximation capabilities of their method is provided. They integrate their approach into an MPNN+Transformer combination, as previously proposed by GraphGPS, replacing the Transformer component with their efficient variant while retaining the same MPNN architectures.

**Strengths:**

1. **Relevance**: The work addresses the critical challenge of developing efficient Transformer variants for graphs.
2. **Motivation**: The study is well-justified and motivated, with clear objectives and potential impacts.
3. **Results**: The authors present compelling results, showing promising improvements in both memory and time efficiency.
4. **Comprehensiveness**: The work covers both theoretical and practical aspects, providing a fairly thorough analysis in each area.

**Weaknesses:**

1. **Clarity and Readability of the Method**
   The method is challenging to follow, especially in certain sections:
   - **Equation 2.2**: It is unclear whether $\mathbf{\alpha}_i$ and $\mathbf{\omega}$ are scalars or vectors. The notations section suggests they are vectors, yet the equation starts with a vector and seems to become scalar. If they are indeed scalars, the connection to the attention mechanism remains unexplained.
   - **Equation 2.4**: This equation introduces several new variable names and vector dimensions without clear definitions, making it difficult to understand. Additionally, the connection to the Transformer architecture is not clearly established in this section.

2. **Connection to Virtual Nodes**
   While the authors’ approach of linking their presentation to virtual nodes is intriguing, it raises a question: does this imply that many underlying theories in this work are already established? For instance, Appendix E in the Exphormer paper [1] includes discussions about virtual nodes that appear to overlap with the concepts in this work.

3. **State-of-the-Art (SoTA) Comparison**
   Although the paper claims to achieve SoTA results across several datasets, it does not compare against models that report better results, such as GRIT [2] or certain optimized results in [3]. For example, paperswithcode provides relevant leaderboard results:
   - [CIFAR10](https://paperswithcode.com/sota/graph-classification-on-cifar10-100k)
   - [MNIST](https://paperswithcode.com/sota/graph-classification-on-mnist)
   - [Pascal-VOC](https://paperswithcode.com/sota/node-classification-on-pascalvoc-sp-1)
   - [COCO-SP](https://paperswithcode.com/sota/node-classification-on-coco-sp)
   In comparison with these benchmarks, the paper’s results do not convincingly indicate SoTA performance.

4. **Graph Edges and Model Efficiency**
   The paper argues that previous methods are inefficient due to the use of graph edges, while their Transformer does not rely on them. However, this advantage becomes less pronounced when the proposed method is combined with the Message Passing Neural Network (MPNN). Therefore, the claim that their method is entirely independent of the number of edges seems somewhat overstated.

-------
[1] Shirzad, H., et al. "Exphormer: Sparse transformers for graphs." *International Conference on Machine Learning*, PMLR, 2023.

[2] Ma, L., et al. "Graph inductive biases in transformers without message passing." *International Conference on Machine Learning*, PMLR, 2023.

[3] Tönshoff, J., et al. "Where did the gap go? Reassessing the long-range graph benchmark." (2023).

**Questions:**

1. Usually, there are parameter constraints on datasets like CIFAR10 and MNIST benchmarks. Does your method meet these parameter constraints? For reference, you can check the constraints outlined in the GraphGPS paper.

2. How does the universal approximation on graphs that considers edges (page 6, lines 270-281) as inputs relate to your method? Your tokens are nodes, which seems to be significantly different from a theory that includes edge information.

3. How can the ability to solve the graph isomorphism problem—which is discrete and not continuous—be inferred from your universal approximation theorems, which are based on continuous function assumptions?

4. What are the connections between this work and linear kernel trick methods such as Nodeformer [1] and Polynormer [2]? The formulations seem very similar in practice.

---------
[1] Wu, Q., et al. "Nodeformer: A scalable graph structure learning transformer for node classification." *Advances in Neural Information Processing Systems* 35 (2022).

[2] Deng, C., et al. "Polynormer: Polynomial-expressive graph transformer in linear time." International Conference on Learning Representations (ICLR) 2024.

---

> ### Author Response · Authors · 2024-11-20
>
> Thank you for your recognition of comprehensiveness and detailed feedback on our work. We address your concerns as below:
>
> **R3.1 Equation clarification.**
>
> We sincerely appreciate your valuable feedback. It is indeed a meaningful suggestion for readers.
>
> - In Eq. (2.2), $\alpha\_j$ is a scalar and $\boldsymbol{\omega}$ is a vector. This equation comes from the original representer theorem, which gives an element-wise explanation of attention output. As you said, for a multi-dimensional output $\tilde{\boldsymbol{g}}$, we indeed need the vector form, naturally generalized from the element-wise definition:   $\boldsymbol{\alpha}\_j\in\mathbb{R}^s$ are vectors, where $j\in[N]$, $N$ is the number of nodes, and the feature mapping $\phi(\boldsymbol{x}):\mathbb{R}^d\rightarrow\mathbb{R}^p$,
> $$
> \tilde{\boldsymbol{g}}=\sum\nolimits\_{j} \boldsymbol{\alpha}\_j\langle\phi(\boldsymbol{x}\_i),\phi(\boldsymbol{x}\_j)\rangle =\sum\nolimits\_{j}  {\rm vec}(\boldsymbol{\alpha}\_j\phi(\boldsymbol{x}\_i)^\top\phi(\boldsymbol{x}\_j)),
> $$
> $$
> \overset{(a)}{=} \sum\nolimits\_{j} \left(\phi(\boldsymbol{x}\_j)^\top\otimes\boldsymbol{\alpha}\_j\right)\phi(\boldsymbol{x}\_i)=\left\langle
> \sum\nolimits\_{j} \phi(\boldsymbol{x}\_j)\otimes\boldsymbol{\alpha}\_j^\top,\phi(\boldsymbol{x}\_i)
> \right\rangle:=\langle\boldsymbol{W}, \phi(\boldsymbol{x}\_i)\rangle,
> $$
> where $\boldsymbol{W}\in\mathbb{R}^{p\times s}$ and $(a)$ comes from the vectorization (${\rm vec}$) property of the [Kronecker product](https://en.wikipedia.org/wiki/Kronecker_product) $\otimes$. We hope these equations could address your concerns.
>
> - Thank you for pointing out this issue. We recognize that the lack of clear definitions for the variables has caused confusion. To address this, we will include clear definitions of the used variables in the Notations part at the beginning of Section 2. Additionally, we will introduce the clear connection to the Transformer architecture in a proper position.
>
> **R3.2 Theorems.**
>
> We would like to first answer the question: the theories in our method has not been already established and we would like to clarify it as follows,
>
> - Theorem 1.
> Our method uses a new primal representation for attention mechanisms, thereby introducing a new primal optimization problem. Theorem 1 establishes the corresponding primal-dual relationship for this new primal representation.
>
> - Theorem 2.
> Given that we establish the primal-dual relationship for our primal representation within the least-squares framework, we are intrigued to ascertain if our method retains the property of a zero-valued objective. Theorem 2 validates this, thereby enabling the application of an alternative optimization approach to address our primal problem.
>
> - Theorem 3.
> In this theorem, we establish that our Primphormer serves as a universal approximator for any permutation-equivariant function. We establish this theorem by firstly demonstrating its representational capacity to Sumformer [1], and subsequently utilizing Sumformer as an intermediary to control the overall error through the application of the triangle inequality (refer to Appendix C.4). A notable distinction between our method and Exphormer[2] pertains to the primal and dual spaces. While Exphormer demonstrated its property in the dual space (sparse attention), our method operates in the primal space. Moreover, our focus centers on exploring worst-case scenarios, employing the supremum norm, in distinction to Exphormer, which utilized the $L_p$ norm to measure the accuracy. These disparities lead to divergent proof strategies.
>
> - Theorem 4.
> Next, we introduced our theorem for any continuous function with positional encodings on compact supports. In contrast, Exphormer [2] necessitated both expander edges and virtual nodes in its sparse attention model. To ensure sufficient node interactions and the universal approximation theorem, [2] required approximately $\mathcal{O}(\log N)$ attention layers, where $N$ denotes the number of nodes. Through our proof, we have streamlined this to a single attention layer by leveraging virtual nodes in our primal representation.
>
> [1] Alberti S, Dern N, Thesing L, et al. Sumformer: Universal approximation for efficient transformers[C]. Topological, Algebraic and Geometric Learning workshops, 2023.
>
> [2] Shirzad H, Velingker A, Venkatachalam B, et al. Exphormer: Sparse transformers for graphs[C]. ICML, 2023.

---

> > ### Author Response · Authors · 2024-11-20
> >
> > **R3.3 SoTA comparison.**
> >
> > We do agree with you that we need to compare against more models. However, it is crucial to note that the main purpose of our approach is to enhance efficiency while maintaining good performance, rather than claiming superiority in terms of accuracy.
> >
> > Following your suggestions, we have conducted experiments comparing our method with two additional approaches [3,4], and the results are reported in Tables 1, 2, and 3. The conclusions drawn align consistently with other reported experiments in the manuscript: overall performance remains stable, with our method demonstrating a notable enhancement in efficiency.
> >
> > Table. 1 Comparison between our method and [3] on the CIFAR10 dataset.
> > | Method | ACC$\uparrow$ | Time(s/epoch)  |    Memory(GB)  |
> > | ------ | ---- | ------- | --- |
> > |    Ours    |   74.13$\pm$0.241   |     32.6    |   2.74  |
> > |    GRIT |   76.46$\pm$0.881   |    158.8    |   22.8  |
> >
> > Table. 2 Comparison between our method and [3] on the MNIST dataset.
> > | Method | ACC$\uparrow$ | Time(s/epoch)  |    Memory(GB)   |
> > | ------ | ---- | ------- | --- |
> > |    Ours    |   98.56$\pm$0.042   |     43.7    |  1.71   |
> > |    GRIT|   98.11$\pm$0.111  |     70.1    |  7.69  |
> >
> > Thank you for suggesting [4], which reported higher F1 scores on the Pascal-VOC and COCO-SP datasets. The difference in performance comes from an additional data preprocessing step (feature normalization, FN), which is parallel to our method and can be implemented similarly. In the following, we report experimental results with and without FN as introduced in [4] in Table 3. Notably, with FN, our method exhibits superior performance.
> >
> >
> > Table. 3 Comparison w/o feature normalization (FN) between our method and [4] on the Pascal-VOC and COCO-SP datasets.
> > |                 | Ours | [4] | Ours+FN | [4]+FN |
> > | ------------------------ | ---- | --- | ------- | ------ |
> > | Pascal-VOC F1$\uparrow$ |   0.3980$\pm$0.0075   |   0.3748$\pm$0.0109  |    0.4602$\pm$0.0077     |    0.4440$\pm$0.0065   |
> > |  COCO-SP   F1$\uparrow$ |   0.3438$\pm$0.0046   |   0.3412$\pm$0.0044  |    0.3903$\pm$0.0061    |   0.3884$\pm$0.0055     |
> >
> > We hope the additional experiments could address your concerns. And we will ensure to add these experiments in the final revision.
> >
> > **R3.4 Model Efficiency.**
> >
> > The question you raised about the entire efficiency of the architecture is very insightful.
> >
> > First, we would like to claim again that the primal representation is independent of the number of edges $|E|$ while dual representation or sparse attention are not, and we did not claim that the entire architecture is independent of $|E|$.
> >
> > Within our model architecture, there are two main computational components: the MPNN and the Transformer. In a worst-case scenario, the overall complexity is upper bounded by the MPNN. However, in practical applications, the proposed primal representation significantly enhances efficiency, as evidenced by our experimental results.
> >
> >
> > [3] Ma L, Lin C, Lim D, et al. Graph inductive biases in transformers without message passing[C]. ICML, 2023.
> >
> > [4] Tönshoff J, Ritzert M, Rosenbluth E, et al. Where did the gap go? reassessing the long-range graph benchmark[J]. arXiv preprint arXiv:2309.00367, 2023.

---

> > > ### Author Response · Authors · 2024-11-20
> > >
> > > **R3.5 Parameter constraints.**
> > >
> > > Yes, we follow the parameter constraints outlined in the GraphGPS paper. We will ensure to add the description in the final revision.
> > >
> > > **R3.6 Dual graph representation.**
> > >
> > > To answer your question, we would like to present the dual graph representation technique introduced in [5]. The original graph can be equivalently transformed into its dual graph, where the edges of the original graph become nodes in the dual graph. Subsequently, we can construct edge Primphormer using input pairs $(i, j, \sigma\_{i,j})$, where $i$ and $j$ represent node indices, and $\sigma\_{i,j}$ denotes the edge indicator.
> > >
> > > **R3.7 Ability to solve graph isomorphism problem.**
> > >
> > > Your question is quite insightful. The concept of universal approximation does not claim that our method can solve the graph isomorphism problem, but rather that it can approximate a solution. It focuses on learning invariant functions within a specified margin of error, which may lead to mislabeling certain graphs. For exactly solving a problem, a method requires not only approximation capability but also representation capability.
> > >
> > > **R3.8 Kernel trick comparison.**
> > >
> > > Thank you for mentioning the two works that also aims at speeding up by matrix decomposition.
> > >
> > > Nodeformer [6] utilizes random features, offering effective approximation when a sufficient number of features are used. Random features differ fundamentally from a kernel trick method, distinguishing them from our approach.
> > >
> > > A notable challenge when employing random features is their heavy reliance on Mercer's condition, which dictates that the kernel must be both symmetric and positive definite. However, an attention matrix is inherently asymmetric. In [7], Polynormer circumvents this issue by incorporating the kernel trick within the activation operation.
> > >
> > > Our method leverages the asymmetric kernel trick, establishing a beneficial primal-dual relationship within our framework that facilitates a explanation of attention mechanisms. Furthermore, the asymmetric kernel trick permits inputs from two distinct spaces (as defined in Definition 1), showing promise as a tool for cross-attention mechanisms, such as image-text attention scores—an ability beyond the scope of [6, 7].
> > >
> > > [5] Anez J, De La Barra T, Pérez B. Dual graph representation of transport networks[J]. Transportation Research Part B: Methodological, 1996, 30(3): 209-216.
> > >
> > > [6] Wu Q, Zhao W, Li Z, et al. Nodeformer: A scalable graph structure learning transformer for node classification[C]. NeurIPS, 2022.
> > >
> > > [7] Deng C, Yue Z, Zhang Z. Polynormer: Polynomial-expressive graph transformer in linear time[C]. ICLR, 2024.

---

> > > > ### Comment · Reviewer_9Dcv · 2024-11-23
> > > >
> > > > I thank the authors for their detailed and thoughtful responses. While the clarifications provided are appreciated, Some of my concerns are still remaining, particularly:
> > > >
> > > > > Theorem 4. Next, we introduced our theorem for any continuous function with positional encodings on compact supports. In contrast, ...
> > > >
> > > > The novelty of this proof is still not clear to me. Regarding the number of layers, are such proofs not generally reliant on the universality of Transformers, which often require an exponentially large number of attention layers? It seems that the inclusion of a $\log(n)$ factor might not substantially differentiate this result. Additionally, there are alternative approaches, such as those used in the BigBird paper [1], which use virtual nodes to prove universality without requiring such assumptions.
> > > >
> > > >
> > > > > We do agree with you that we need to compare against more models. However, it is crucial to note that the main purpose of our approach is to enhance efficiency while maintaining good performance, rather than claiming superiority in terms of accuracy.
> > > >
> > > > I understand that efficiency-focused methods may not always achieve SoTA results and that there is often a trade-off between efficiency and accuracy. However, it is important for the paper to present its claims with precision. Announcing SoTA results without verifying them against the current benchmarks can be misleading. While I appreciate the inclusion of new experimental results, I would encourage the authors to acknowledge any prior overstatements and ensure their claims are appropriately refined to reflect their findings.
> > > >
> > > >
> > > > > R3.4 Model Efficiency.
> > > >
> > > > While I agree that the Transformer component of your method is independent of $|E|$, I still find the paper's discussion of computational complexity somewhat unclear. Your method seems to achieve good results primarily in conjunction with MPNNs. Given this, the importance of the $O(|V|)$ complexity of the Transformer part remains ambiguous, especially considering that the MPNN component—if it follows the GraphGPS approach—uses a customized GatedGCN, which can be more computationally expensive than the self-attention mechanism itself.
> > > >
> > > >
> > > > [1] Zaheer, Manzil, et al. "Big Bird: Transformers for longer sequences." Advances in Neural Information Processing Systems 33 (2020).

---

> > > > > ### Author Response · Authors · 2024-11-24
> > > > >
> > > > > Thank you very much for providing valuable feedback.
> > > > >
> > > > > **R3.9 Discussion on Theorem 4**
> > > > >
> > > > > In Theorem 4, our network architecture differs from that of Exphormer and Bigbird, thereby we establish a new theorem not previously addressed in previous works. The proof of Theorem 4 adopts a different strategy compared to earlier works, and we would like to summarize it as follows:
> > > > >
> > > > > 1. We use a different technique from previous works. While Exphormer and Bigbird rely on the concept of contextual mappings, necessitating an exponentially large number of attention layers in their Transformers, our proof demonstrates that our architecture only needs one attention layer to represent Sumformer, and subsequently utilizes Sumformer as an intermediary to control the overall error through the application of the triangle inequality (refer to Appendix C.4 and C.5).
> > > > > 2. Our focus lies on worst-case scenarios utilizing the supremum norm, contrasting Exphormer's utilization of the $L_p$ norm where $1\le p<\infty$ to measure accuracy.
> > > > >
> > > > > **R3.10 Refined Claims**
> > > > >
> > > > > Thank you for your valuable suggestion. We acknowledge the importance of presenting precise claims and recognize that the current statements may lead to misunderstandings. Following your recommendations, we have reviewed our manuscript and identified 3 sentences referencing "SoTA results" (page 2, line 68; page 7, line 327; page 8, line 409). We will ensure to adjust them accordingly in the final revision:
> > > > >
> > > > > >Page 2 line 68: Through extensive experiments on various graph benchmarks, we show that Primphormer achieves competitive empirical results while maintaining a more user-friendly memory and computational costs.
> > > > >
> > > > > >Page 7 line 327: It is observed that Primphormer outperforms on MNIST and ranks as the second-best performer on two additional datasets, showcasing its strong performance across various dataset types.
> > > > >
> > > > > >Page 8 line 409: In summary, our experiments demonstrate that Primphormer exhibits competitive performance while maintaining user-friendly memory and computational costs.
> > > > >
> > > > >
> > > > > **R3.11 Model efficiency**
> > > > >
> > > > > We follow the GraphGPS approach and use a customized GatedGCN. Here, we provide memory cost and time for GatedGCN and standard Transformer and our primal representation modules on the MalNet-Tiny dataset within the identical experimental setup outlined in the manuscript.
> > > > >
> > > > > | MalNet-Tiny | Time(s/epoch) | Memory(GB) |
> > > > > | ----------- | ------------- | ---------- |
> > > > > | GatedGCN    | 24.5          | 1.92       |
> > > > > | Transformer | 197.9         | 32.4       |
> > > > > | Ours | 48.6          | 2.22       |
> > > > >
> > > > >
> > > > > For a middle-sized graph dataset MalNet-Tiny, it is evident that the Transformer module consumes more computational and memory resources, highlighting the necessity of modifying Transformer modules. We hope this could address your concerns.

---

> > > > > > ### Comment · Reviewer_9Dcv · 2024-11-25
> > > > > >
> > > > > > I want to thank the authors for their detailed responses. Since most of my concerns have been addressed, I would like to raise my score to 6.

---

> > > > > > > ### Author Response · Authors · 2024-11-25
> > > > > > >
> > > > > > > We sincerely appreciate your efforts in re-evaluating our work. We have incorporated the discussion in the manuscript.

---

### Official Review · Reviewer_znmU · 2024-11-04

**Soundness:** 4
**Presentation:** 4
**Contribution:** 4
**Rating:** 6
**Confidence:** 3

**Summary:**

This paper introduces Primphormer, which reduces computational complexity from quadratic to linear by representing self-attention as a dual representation and modeling it in primal space.

**Strengths:**

The idea is interesting and it reduces the time complexity using primal space.

The authors provide clear pseudocode and detailed implementation guidelines, making the work practical for real-world applications.


The experimental evaluation is comprehensive, including lots of datasets from different domains.

**Weaknesses:**

1. Using virtual nodes could potentially bring bottlenecks in information flow for graphs with complex hierarchical structures or when important information needs to be preserved across distant nodes.


2. The transition to primal space requires specific mathematical conditions, such as accommodating the inherent asymmetry of attention scores, which limits its applicability.

**Questions:**

None

---

> ### Author Response · Authors · 2024-11-20
>
> Thank you very much for finding our idea interesting and providing insightful comments. We also appreciate your recognition of the practicality of our work. We address your concerns as below:
>
> **R2.1 Information flow via virtual nodes.**
>
> Virtual nodes (VNs) create shortcuts for sharing information between graph nodes, facilitating global information exchange and enhancing information flow, as shown in some previous works [1,2,3]. In cases where important information needs to be preserved across distant nodes, VNs can improve information flow.
>
> Your suggestion about complex hierarchical structures hits the idea of a very recent paper [4] that introduces an extension of VNs to enhance information exchange. We believe this is an interesting and important scenario. This technique could also be integrated into our method, which we plan to explore in future work.
>
>
> **R2.2 Applicability of our method.**
>
>
> The transition to the primal space does not require any preconditions. Regarding your question, the discussion about the asymmetric kernel actually covers the symmetric one. We hope this clarification could address your concern: Primphormer can handle both asymmetric and symmetric attension score matrices.
>
> We will make the necessary modifications in the manuscript to address the issues you have pointed out in the manuscript.
>
> [1] Hu W, Fey M, Zitnik M, et al. Open graph benchmark: Datasets for machine learning on graphs[C]. NeurIPS, 2020.
>
> [2] Hwang E J, Thost V, Dasgupta S S, et al. An analysis of virtual nodes in graph neural networks for link prediction[C]. LoG, 2022.
>
> [3] Cai C, Hy T S, Yu R, et al. On the connection between mpnn and graph transformer[C]. ICML, 2023.
>
> [4] Vonessen C, Grötschla F, Wattenhofer R. Next level message-passing with hierarchical support graphs[J]. arXiv preprint arXiv:2406.15852, 2024.

---

> > ### Comment · Reviewer_znmU · 2024-11-26
> >
> > Thanks for your reply, especially on the questions about the ability to handle both asymmetric and symmetric attention score matrices. My concerns have been solved.

---

### Official Review · Reviewer_gkyd · 2024-11-04

**Soundness:** 2
**Presentation:** 1
**Contribution:** 2
**Rating:** 3
**Confidence:** 4

**Summary:**

This paper proposes an efficient graph Transformer model using an asymmetric kernel trick. Specifically, the model does not need to compute pair-wise scores, so there is no extra computational burden. The key analysis of this model is based on (or, say, similar to) [1], which reformulates the original problem to a dual problem. This primal-dual approach leverages the graph information to adjust the basis of outputs and has more expressive power. Furthermore, the authors prove that the proposed model, namely Primphormer, could be a good universal approximator for arbitrary continuous functions. Experimental results also show the proposed model has better performance while using less memory and computational costs.

**Strengths:**

1. The formulation of primal graph Transformer algorithm to dual is interesting. The dual problem gives a nice solution via KKT condition. The primal-dual formulation gives some nice theoretical properties.

2. The experimental results look promising. Compared with current state-of-the-art method, the proposed mehthods have better performance over all while using less memory and computation resources.

**Weaknesses:**

In general, the paper proposes a new method for graph presentation learning. The experimental results look promising. However, I found this paper is heavily based on a previous work (see [1]). Hence, the overall novelty is very limited. Some weaknessnes are listed as follows:

1. Concern about the definition of primal problem: The formulation of original problem of graph Transformer is defined as in (2.4). Why is this definition is the right one?

2. Concern about the overal novelty of this paper: The formulation of (2.4) is very similar to the formulation used in [1]. I would believe that the theorems and dual formulation will largely follow the techniques used in [1]. If not, please explain what are the differences between these two. At this point, the overall novelty of this paper is limited.

3. Some definition is missing citation: The Definition of (2.4) is very similar to Definition 2.1 of [1]. It would be more helpful if the authors put citation here as the definition is not original.

4. Difference between Theorem 4 and Lemma 4.2 in [1]. I found a large context of this Theorem and Lemma 4.2 in [1] is quite similar. Please explain more on the difference between these two.

[1] Yingyi Chen, Qinghua Tao, Francesco Tonin, and Johan A. K. Suykens. Primal-attention: Selfattention through asymmetric kernel SVD in primal representation. In the Thirty-seventh Conference on Neural Information Processing Systems, 2023.

**Questions:**

See the weakness section.

---

> ### Author Response · Authors · 2024-11-17
>
> Thank you for your careful reading and insightful suggestions. We address your concerns as below:
>
> **R1.1 & R1.2 Novelty.**
>
> The key to accelerating the attention from the primal-dual perspective is to find good approximation for $o$:
> $o(\boldsymbol{x})=\sum\nolimits\_{i} v(\boldsymbol{x}\_i)\kappa(\boldsymbol{x}, \boldsymbol{x}\_i)=\sum\nolimits\_{i} v(\boldsymbol{x}\_i)\langle\phi\_q(\boldsymbol{x}), \phi\_k(\boldsymbol{x}\_i)\rangle$.
> Here, $v(x_i) \in \mathbb{R}^{d_o}$ provides the basis and $\kappa(x,x_i) \in \mathbb{R}$ are the weight in the corresponding basis. The approximation error depends on both basis and weights, but the effect of basis is more significant.
>
> To achieve better approximation, there are two ways to introduce data:
> * Data Adaptive Weight
> this is the way of [1]: the weight is set to be $\langle f\_X\phi\_q(\boldsymbol{x}), f\_X\phi\_k(\boldsymbol{y})\rangle$
>
> * Data Adaptive Basis
> this is the way proposed in our paper: the basis is set to be $F\_X\boldsymbol{h}\_{e}$,   $F\_X\boldsymbol{h}\_{r}$, where $\boldsymbol{h}\_{e}, \boldsymbol{h}\_{r}$ are the dual variables.
>
> We hope this comparision could highlight the fundemental difference between the two papers. The advantage of making the basis data-adaptive is evident from the universal approximation capability (Theorem 3 and Theorem 4). Only when sufficient flexibility is introduced, we could prove such theorems. Additional evidence comes from the numerical experiments, see Table 3 for Primphormer (ours) vs Prim-atten ([1]).
>
> The similarity in formulation comes from the fact that both [1] and our method are built on the same least square framework [2], where the dual variables are from equation constraints, i.e., Eq. (6) in [1] and Eq. (2.4) in our paper.
>
> Lastly, the design of data-dependent projection $f\_X$ is also important. In this paper, we use a different, data-adaptive scheme:
>
> |  | ours | [1] |
> | -------- | -------- | -------- |
> | $f\_X$     | $\boldsymbol{F}+X\boldsymbol{1}\boldsymbol{1}^\top$     | Uniform and ordered sampling   |
>
> The $f_X$ that we used can also be found in [3, 4] and is known as permutation-equivariant projection in deepsets. As we discussed in the manuscript, graph structures are determined by edges and the arrangement or ordering of nodes is not explicitly specified. Therefore, this formulation is more suitable for our tasks.
>
>
> **R1.3 Definition Citations.**
>
> In fact, we have cited several papers that use this definition in the line above Definition 1. But we do agree with you that using an inline citation may be better. We will certainly modify it in the final version.
>
>
> **R1.4 Theorems between two works.**
>
> The theorems of the zero-valued objective are quite important for our method and the one in [1]. This property is essential for making the alternative optimization approach applicable to solving the primal problem.
>
> As explained before, both of them are in the same least square framework so that the proofs of Theorem 2 and Lemma 4.2 in [1] are quite similar (actually, they are both following Corollary 1 in [2]). For our new method, we still need to prove it, although there is little technical contribution in the proof and we did not list this as a contribution.
>
> Actually, the universal approximation  (Theorem 3 and Theorem 4) is the main property we would like to highlight. Our Primphormer's ability to approximate any continuous function on a compact domain (see Appendix C.4 and C.5) is not found in the architecture of [1]. This is a significant advantage of our new representation, which introduces a data-adaptive basis, as explained in R1.1 and R1.2.
>
>
> [1] Chen Y, Tao Q, Tonin F, et al. Primal-attention: Self-attention through asymmetric kernel svd in primal representation[C]. NeurIPS, 2024.
>
> [2] Suykens J. A. K. SVD revisited: A new variational principle, compatible feature maps and nonlinear extensions[J]. ACHA, 2016, 40(3): 600-609.
>
> [3] Zaheer M, Kottur S, Ravanbakhsh S, et al. Deep sets[C]. NeurIPS, 2017.
>
> [4] Cai C, Hy T S, Yu R, et al. On the connection between mpnn and graph transformer[C]. ICML, 2023.

---

> > ### Comment · Reviewer_gkyd · 2024-11-25
> >
> > I thank the authors for providing a detailed explanation. I also read other reviewers' comments. It seems that my concerns are unique.
> >
> > 1) From the author's response, the Data Adaptive Basis has the advantage over the Data Adaptive Weight method is because the universal approximation capability holds for yours but not for the Data Adaptive Weight method? I thought the universal approximation capability comes from the attention mechanism itself. If this is the case the authors may prove that data adaptive weight way cannot have universal approximation capability.
> >
> > 2) I do think the citation in the definition itself is important because otherwise it could be misleading readers to think the definition is novel.
> >
> > 3) "For our new method, we still need to prove it, although there is little technical contribution in the proof and we did not list this as a contribution." I thank the authors make this statement. I also agree that the technical contribution is not main goal of this paper.
> >
> > Currently, I will keep my score unchanged and decide to make final decision during reviewer's discussion period.

---

> > > ### Author Response · Authors · 2024-11-25
> > >
> > > Thank you for the insightful discussion.
> > >
> > > While the standard attention mechanism possesses the universal approximation property, employing a primal representation to lighten and approximate the attention mechanism in the primal space introduces a different network architecture and potentially reduces the capabilities of the attention mechanism. Therefore, a crucial question arises: Can the universal approximation property be preserved in this context? For Primphormer, we have demonstrated this, not by relying on the standard attention mechanism, but by showcasing Primphormer's representation capabilities to Sumformer [1].
> > >
> > > Regarding the advantages of data-adaptive bases over data-adaptive weights, these represent two different approaches that are hard to compare directly. Theoretically, we have proven the universal approximation property, while the data-adaptive-weight scheme has not yet found it (but we cannot say there is no such property since disapproving something is always hard). Experimentally, Primphormer has shown significant improvements (refer to Table 3 for a comparison between "Primphormer" and "Prim-Atten": +2.5% in CIFAR-10, +0.7% in MalNet-tiny, +8.1% in PascalVOC-SP, +1.7% in Peptides-Func, +0.4% in OGBN-products)
> > >
> > > [1] Alberti S, Dern N, Thesing L, et al. Sumformer: Universal approximation for efficient transformers[C]. Topological, Algebraic and Geometric Learning workshops, 2023.

---

### Comment · Area_Chair_QTre · 2024-11-23
**Reminder: Please Review Author Responses**

Dear Reviewers,

As the discussion period is coming to a close, please take a moment to review the authors’ responses if you haven’t done so already. Even if you decide not to update your evaluation, kindly confirm that you have reviewed the responses and that they do not change your assessment.

Thank you for your time and effort!

Best regards,
AC

---

### Meta-Review · Area_Chair_QTre · 2024-12-12

**Metareview:**

The paper introduces a graph Transformer that leverages the feature representation induced by an asymmetric kernel trick. This approach purportedly reduces computational complexity from quadratic to linear, enhances efficiency, and maintains theoretical guarantees such as universal approximation. Experimental results indicate improvements in memory usage and computational cost compared to state-of-the-art methods. The theoretical framework connects to the primal-dual problem reformulation, offering an innovative perspective.

**Strengths**

The primal representation induced by an asymmetric kernel introduces a new approach to improving Transformer efficiency.  The paper proves the universal approximation properties of the proposed method. Extensive experiments across diverse datasets demonstrate the paper's claims about memory and time efficiency improvements.

**Weaknesses**

The paper builds upon prior techniques (e.g., Primal-Attention, Exphormer). In particular, the authors are suggested to compare the proposed methods with existing graph transformers. There are also some presentation issues, e.g., the reason to achieve universal approximation, and the relation between the primal-dual setting and the implementation. Some baselines and experiments are also suggested to be added.

Overall, I lean towards a rejection, as some concerns about the novelty, and presentation issues, are not sufficiently addressed. Moreover, I suggest the authors compare the method with many graph models of linear complexity [1,2] to achieve more solid arguments and address the concerns of novelty.

[1] NodeFormer: A Scalable Graph Structure Learning Transformer for Node Classification, Wu et al., NeurIPS 2022
[2] What Can We Learn from State Space Models for Machine Learning on Graphs? Huang et al., arxiv 2024

**Additional Comments On Reviewer Discussion:**

The authors addressed some concerns, including missing evaluations, complexity analysis, questions about virtual nodes, and the potentially limited applications of the proposed method. While the authors proposed steps to address these issues, the concerns regarding the novelty and presentation of the paper remain inadequately addressed, leaving reviewers unconvinced.

---

### Decision · Program_Chairs · 2025-01-22

Reject